# Prior-itizing Privacy: A Bayesian Approach to Setting the Privacy Budget in Differential Privacy

**Zeki Kazan**
Department of Statistical Science
Duke University
Durham, NC 27708
zekican.kazan@duke.edu

**Jerome P. Reiter**
Department of Statistical Science
Duke University
Durham, NC 27708
jreiter@duke.edu

## Abstract

When releasing outputs from confidential data, agencies need to balance the analytical usefulness of the released data with the obligation to protect data subjects' confidentiality. For releases satisfying differential privacy, this balance is reflected by the privacy budget, $\varepsilon$. We provide a framework for setting $\varepsilon$ based on its relationship with Bayesian posterior probabilities of disclosure. The agency responsible for the data release decides how much posterior risk it is willing to accept at various levels of prior risk, which implies a unique $\varepsilon$. Agencies can evaluate different risk profiles to determine one that leads to an acceptable trade-off in risk and utility.

## 1 Introduction

Differential privacy (DP) [11] is a gold standard definition of what it means for data curators, henceforth referred to as agencies, to protect individuals' confidentiality when releasing sensitive data. The confidentiality guarantee of DP is determined principally by a parameter typically referred to as the privacy budget $\varepsilon$. Smaller values of $\varepsilon$ generally imply greater confidentiality protection at the cost of injecting more noise into the released data. Thus, agencies must choose $\varepsilon$ to balance confidentiality protection with analytical usefulness. This balancing act has resulted in a wide range of values of $\varepsilon$ in practice. For example, early advice in the field recommends $\varepsilon$ of "0.01, 0.1, or in some cases, $\log(2)$ or $\log(3)$" [10], whereas recent large-scale implementations use $\varepsilon = 8.6$ in OnTheMap [31], $\varepsilon = 14$ in Apple's use of local DP for iOS 10.1.1 [44], and an equivalent of $\varepsilon = 17.14$ in the 2020 decennial census redistricting data release [1].

Some decision makers may find interpreting and selecting $\varepsilon$ difficult [20, 24, 29, 36, 43]. For example, a recent study of practitioners using DP Creator [43] found that users wished for more explanation about how to select privacy parameters and better understanding of the effects of this choice. One potential path for providing guidance is to convert the task of selecting $\varepsilon$ to setting bounds on the allowable probabilities of adversaries learning sensitive information from the released data, as in the classical disclosure limitation literature [8, 14, 41]. There is precedent for this approach: prior work in the literature suggests that "privacy semantics in terms of Bayesian inference can shed more light on a privacy definition than privacy semantics in terms of indistinguishable pairs of neighboring databases" [28] and prevailing advice for setting $\delta$ in $(\varepsilon, \delta)$-DP originates from such Bayesian semantics [25].

We propose that agencies utilize relationships between DP and Bayesian semantics [2, 11, 25, 26, 27, 28, 29, 33] to select values of $\varepsilon$ that accord with their desired confidentiality guarantees. The basic idea is as follows. First, the agency constructs a function that summarizes the maximum posterior probability of disclosure permitted for any prior probability of disclosure. For example, for a prior risk of 0.001, the agency may be comfortable with a ten-fold (or more) increase in the ratio of posterior risk to prior risk, whereas for a prior risk of 0.4, the agency may require the increase not exceed, say, 1.2. Second, for each prior risk value, the agency converts the posterior-to-prior ratio into the largest

$\varepsilon$ that still ensures the ratio is satisfied. Third, the agency selects the smallest $\varepsilon$ among these values, using that value for the data release. Importantly, the agency does not use the confidential data in these computations—they are theoretical and data free—so that they do not use up part of the overall privacy budget. Our main contributions include:

- We propose a framework for selecting $\varepsilon$ under certain conditions (see Section 3) that applies to any DP mechanism, does not use additional privacy budget, and can account for disclosure risk from both an individual's inclusion and the sensitivity of values in the data.

- We enable agencies to tune the choice of $\varepsilon$ to achieve their desired posterior-to-prior risk profile. This can avoid setting $\varepsilon$ unnecessarily small if, for example, the agency tolerates larger posterior-to-prior ratios for certain prior risks. In turn, this can help agencies better manage trade-offs in disclosure risk and data utility.

- We give theoretic justification for the framework and derive closed-form solutions for the $\varepsilon$ implied by several risk profiles. For more complex risk profiles, we also provide a general form for $\varepsilon$ as a minimization problem.

To streamline the discussion, we focus on the release of discrete-valued statistics computed on discrete-valued data. Extension to continuous-valued statistics and data can be accomplished by replacing sums with integrals and PMFs with PDFs throughout the theorem statements and proofs.

## 2  Background and Motivation

We first describe some aspects of DP and Bayesian probabilities of disclosure relevant for our approach. We summarize all notation and definitions in Tables 2 and 3 in Appendix A.1.

### 2.1  Differential Privacy

Let $\mathbf{P}$ represent a population of individuals. The agency has a subset of $\mathbf{P}$, which we call $\mathbf{Y}$, comprising $n$ individuals measured on $d$ variables. For any individual $i$, let $Y_i$ be the length-$d$ vector of values corresponding to individual $i$, and let $I_i = 1$ when individual $i$ is in $\mathbf{Y}$ and $I_i = 0$ otherwise. For all $i$ such that $I_i = 1$, let $\mathbf{Y}_{-i}$ be the $(n-1) \times d$ matrix of values for the $n-1$ individuals in $\mathbf{Y}$ excluding individual $i$.[1] The agency wishes to release some function of the data, $T(\mathbf{Y})$. We assume $\mathbf{Y}$ and $T(\mathbf{Y})$ each have discrete support but may be many-dimensional. The agency turns to DP and will release $T^*(\mathbf{Y})$, a noisy version of $T(\mathbf{Y})$ under $\varepsilon$-DP.

Our work focuses on unbounded DP as defined in [9], which involves the removal or addition of one individual's information.[2] If a function $T^*(\mathbf{Y})$ with discrete support satisfies unbounded $\varepsilon$-DP, then for all $i$, all $y$ in the support of $Y_i$, all $\mathbf{y}_{-i}$ in the support of $\mathbf{Y}_{-i}$, and all $t^*$ in the support of $T^*(\mathbf{Y})$,

$$e^{-\varepsilon} \leq \frac{P[T^*(\mathbf{Y}) = t^* \mid Y_i = y, I_i = 1, \mathbf{Y}_{-i} = \mathbf{y}_{-i}]}{P[T^*(\mathbf{Y}_{-i}) = t^* \mid I_i = 0, \mathbf{Y}_{-i} = \mathbf{y}_{-i}]} \leq e^{\varepsilon}. \tag{1}$$

In settings where the statistic of interest is a count, a commonly used algorithm to satisfy DP is the geometric mechanism [18].

**Definition 1** (Geometric Mechanism). *Let $T(\mathbf{Y}) \in \mathbb{Z}$ be a count statistic and suppose we wish to release a noisy count $T^*(\mathbf{Y}) \in \mathbb{Z}$ satisfying $\varepsilon$-DP. The geometric mechanism produces a count centered at $T(\mathbf{Y})$ with noise from a two-sided geometric distribution with parameter $e^{-\varepsilon}$. That is,*

$$P[T^*(\mathbf{Y}) = t^* \mid T(\mathbf{Y}) = t] = \frac{1 - e^{-\varepsilon}}{1 + e^{-\varepsilon}} e^{-\varepsilon|t^* - t|}, \qquad t^* \in \mathbb{Z}. \tag{2}$$

Under the geometric mechanism, the variance of $T^*(\mathbf{Y})$ is $2e^{-\varepsilon}/(1 - e^{-\varepsilon})^2$. The variance increases as $\varepsilon$ decreases, reflecting the potential loss in data usefulness when setting $\varepsilon$ to a small value.

Of course, data usefulness is only one side of the coin. Agencies also need to assess the implications of the choice of $\varepsilon$ for disclosure risks [7, 42, 43]. Prior works on setting $\varepsilon$ focus on settings where (i)

---

[1]For all $i$ such that $I_i = 0$ we let $\mathbf{Y}_{-i}$ be an $(n-1) \times d$ matrix of individuals not including individual $i$.

[2]This is in contrast to bounded DP [11], which involves the change of one individual's information.

the data have yet to be collected, and the goal is to simultaneously select $\varepsilon$ and determine how much to compensate individuals for their loss in privacy [6, 15, 23, 30], (ii) the population is already public information, and the goal is to protect which subset of individuals is included in a release [29, 35], or (iii) data holders can utilize representative test data or the confidential data set itself [4, 17, 20, 35]. We focus on the common setting where data already have been collected, the population they are drawn from is not public information, and no data are available for tuning $\varepsilon$.

## 2.2 Bayesian Measures of Disclosure Risk

Consider an adversary who desires to learn about some particular individual $i$ in $\mathbf{Y}$ using the release of $T^*(\mathbf{Y})$. We suppose that the adversary has a model, $\mathcal{M}$, for making predictions about the components of $Y_i$ that they do not already know.[3] For example, the adversary could know some demographic information about individual $i$ but not some other sensitive variable. The adversary might predict this sensitive variable from the demographic information using a model estimated with proprietary information or data from sources like administrative records. We assume that the release mechanism for $T^*(\mathbf{Y})$ is known to the adversary, that the DP release mechanism does not depend on $\mathcal{M}$, and that, under $\mathcal{M}$, the observations are independent but not necessarily identically distributed. These conditions are formalized in Section 3. For our ultimate purpose, i.e., helping agencies set $\varepsilon$, the exact form of the adversary's $\mathcal{M}$ is immaterial. In fact, as we shall discuss, we are not concerned whether the adversary's predictions from $\mathcal{M}$ are highly accurate or completely awful.

On a technical note, we make the distinction that the agency views $\mathbf{Y}$ and $I_i$ as fixed quantities, since it knows which rows are in the collected data and what values are associated to each row. The adversary, however, views $\mathbf{Y}$ and $I_i$ as random variables, and thus probabilistic statements about these quantities are well defined from the adversary's perspective. Notationally, we signify that a probabilistic statement is from the adversary's perspective via the subscript $\mathcal{M}$.

Let $\mathcal{S}$ be the subset of the support of $Y_i$ that the agency considers a privacy violation. For example, if $d = 1$ and $\mathbf{Y}$ is income data, then $\mathcal{S}$ may be the set of possible incomes within 5,000 or within 5% of the true income for individual $i$. Alternatively, if $d = 1$ and $\mathbf{Y}$ is binary, then $\mathcal{S}$ is a subset of $\{0, 1\}$. The selection of $\mathcal{S}$ must not depend on $\mathbf{P}$, as this might constitute a privacy violation.

The agency may be concerned about the risk that the adversary determines individual $i$ is in $\mathbf{Y}$ or the risk that the adversary makes a disclosure for individual $i$; that is, $I_i = 1$ and $Y_i \in \mathcal{S}$, respectively. Assuming that the adversary's model puts nonzero probability mass on these events, we can express their relevant prior probabilities as follows.

$$P_{\mathcal{M}}[I_i = 1] = p_i, \qquad P_{\mathcal{M}}[Y_i \in \mathcal{S} \mid I_i = 1] = q_i. \tag{3}$$

For fixed $p_i$ and $q_i$, we can measure the risk of disclosure for individual $i$ in a number of ways. Drawing from [33], one measure is the relative disclosure risk, $r_i(p_i, q_i, t^*)$. Writing the noisy statistic as $T^*$ and suppressing the dependence on $\mathbf{Y}$ or $\mathbf{Y}_{-i}$, this is defined as follows.

**Definition 2** (Relative Disclosure Risk). *For fixed data $\mathbf{Y}$, individual $i$, adversary's model $\mathcal{M}$, and released $T^* = t^*$, the relative disclosure risk is the posterior-to-prior risk ratio,*

$$r_i(p_i, q_i, t^*) = \frac{P_{\mathcal{M}}[Y_i \in \mathcal{S}, I_i = 1 \mid T^* = t^*]}{P_{\mathcal{M}}[Y_i \in \mathcal{S}, I_i = 1]}. \tag{4}$$

The relative risk can be decomposed into the posterior-to-prior ratio from inclusion ($I_i$) and the posterior-to-prior ratio from the values ($Y_i$). We have

$$r_i(p_i, q_i, t^*) = \frac{P_{\mathcal{M}}[Y_i \in \mathcal{S} \mid I_i = 1, T^* = t^*]}{P_{\mathcal{M}}[Y_i \in \mathcal{S} \mid I_i = 1]} \cdot \frac{P_{\mathcal{M}}[I_i = 1 \mid T^* = t^*]}{P_{\mathcal{M}}[I_i = 1]}. \tag{5}$$

The relative risk, however, does not tell the full story. As discussed in [22], the data holder also may care about absolute disclosure risks, $a_i(p_i, q_i, t^*)$.

**Definition 3** (Absolute Disclosure Risk). *For fixed data $\mathbf{Y}$, individual $i$, adversary's model $\mathcal{M}$, and released $T^* = t^*$, the absolute disclosure risk is the posterior probability,*

$$a_i(p_i, q_i, t^*) = P_{\mathcal{M}}[Y_i \in \mathcal{S}, I_i = 1 \mid T^* = t^*]. \tag{6}$$

Since $r_i(p_i, q_i, t^*) = a_i(p_i, q_i, t^*)/(p_i q_i)$, we can convert between these risk measures.

---

[3]To be technically precise, we suppose the adversary assigns zero probability to predictions of $Y_i$ with values of components that they know to be incorrect.

## 2.3 Risk Profiles

The quantities from Section 2.2 can inform the choice of $\varepsilon$. For example, DP implies that

$$r_i(p_i, q_i, t^*) \leq e^{2\varepsilon} \tag{7}$$

for all $(p_i, q_i, t^*)$. [4] The inequality in (7) implies a naive strategy for setting $\varepsilon$: select a desired bound on the relative risk, $r^*$, and set $\varepsilon = \log(r^*)/2$. Practically, however, this strategy suffers from two drawbacks that could result in a smaller recommended $\varepsilon$ than necessary. First, for any particular $p_i$ and $q_i$, the bound in (7) need not be tight. In fact, this bound is actually quite loose across a wide range of values. Second, this strategy does not account for agencies willing to tolerate different relative risks for different prior probabilities. For example, if $p_i q_i = 0.25$, an agency may wish to limit the adversary's posterior to $a_i(p_i, q_i, t^*) \leq 2 \times 0.25 = 0.5$, but for $p_i q_i = 10^{-6}$, the same agency may find a limit of $a_i(p_i, q_i, t^*) \leq 2 \times 10^{-6}$ unnecessarily restrictive.

Rather than restricting to a constant bound, the agency can consider tolerable relative risks as a function of a hypothetical adversary's prior probabilities. We refer to this function as the agency's risk profile, denoted $r^*(p_i, q_i)$. Thus, the agency establishes a $r^*(p_i, q_i)$ so that, for all $p_i$, $q_i$, and $t^*$,

$$r_i(p_i, q_i, t^*) \leq r^*(p_i, q_i). \tag{8}$$

As the following results show, the requirement in (8) translates to a maximum value of $\varepsilon$.

# 3 Theoretical Results

In this section, we describe our main theoretical results. To devote more space to examples and applications, we describe supporting results in Appendix B.1 and provide all proofs in Appendix E.

We begin by formalizing the assumptions on the release mechanism for $T^*$ and the adversary's model, $\mathcal{M}$. We emphasize that $P[T^*(\mathbf{Y}) = t^* \mid Y_i = y, \mathbf{Y}_{-i} = \mathbf{y}_{-i}, I_i = 1]$ and $P[T^*(\mathbf{Y}_{-i}) = t^* \mid \mathbf{Y}_{-i} = \mathbf{y}_{-i}, I_i = 0]$ are probabilities under the actual DP mechanism used for the release of $T^*(\mathbf{Y})$ when $i$ is and is not included, respectively. We use these probabilities in two assumptions as follows.

**Assumption 1.** *For all $y$ in the support of $Y_i$, $\mathbf{y}_{-i}$ in the support of $\mathbf{Y}_{-i}$, and $t^*$ in the support of $T^*$,*

$$P_\mathcal{M}[T^*(\mathbf{Y}) = t^* \mid Y_i = y, \mathbf{Y}_{-i} = \mathbf{y}_{-i}, I_i = 1] = P[T^*(\mathbf{Y}) = t^* \mid Y_i = y, \mathbf{Y}_{-i} = \mathbf{y}_{-i}, I_i = 1]$$
$$P_\mathcal{M}[T^*(\mathbf{Y}_{-i}) = t^* \mid \mathbf{Y}_{-i} = \mathbf{y}_{-i}, I_i = 0] = P[T^*(\mathbf{Y}_{-i}) = t^* \mid \mathbf{Y}_{-i} = \mathbf{y}_{-i}, I_i = 0].$$

Assumption 1 implies that the mechanism for releasing $T^*$ given $\mathbf{Y}$ is known to the adversary and that the adversary uses the actual release mechanism. An adversary who uses something other than the actual release mechanism to compute probabilities is unlikely in general to find more success than the adversary who does. Hence, Assumption 1 assures that the agency's computations are principled and cover a type of "worst-case" analysis.

**Assumption 2.** *For all $y$ in the support of $Y_i$ and $\mathbf{y}_{-i}$ in the support of $\mathbf{Y}_{-i}$,*

$$P_\mathcal{M}[\mathbf{Y}_{-i} = \mathbf{y}_{-i} \mid I_i = 1, Y_i = y] = P_\mathcal{M}[\mathbf{Y}_{-i} = \mathbf{y}_{-i} \mid I_i = 0]. \tag{9}$$

Assumption 2 implies that the adversary's beliefs about the distribution of $\mathbf{Y}_{-i}$ do not change whether individual $i$ is included in the data or not, nor do they depend on individual $i$'s confidential values. This assumption is similar to one explored in [28], who consider the case where "the presence/absence/record-value of each individual is independent of the presence/absence/record-values of other individuals," although we do not require that (9) holds for all $i$. Since DP is designed to capture the effect that a single atomic unit of data has on the output distribution, assuming this type of independence structure ensures one atomic unit is a single data point.[5] We believe these two assumptions are weaker than those in related work on choosing $\varepsilon$, which we discuss in Section 6.

Under Assumption 1 and Assumption 2, we can relate $\varepsilon$ to the distribution of $T^*$ unconditional on $\mathbf{Y}_{-i}$ via the following lemma. This result is similar, but not identical to, Theorem 6.1 in [28].

---

[4]This is proved in [19] under bounded DP in the case where $|\mathcal{S}| = 1$. We show in Corollary 1 in the appendix that, under our assumptions, it holds for larger sets and for unbounded DP (with $\varepsilon$ replaced by $2\varepsilon$).

[5]We thank a referee for highlighting this rationale for Assumption 2.

**Lemma 1.** *Under Assumption 1 and Assumption 2, if the release of $T^* = t^*$ satisfies $\varepsilon$-DP, then for any subset $\mathcal{S}$ of the domain of $Y_i$, we have*

$$e^{-\varepsilon} \leq \frac{P_{\mathcal{M}}[T^* = t^* \mid Y_i \in \mathcal{S}, I_i = 1]}{P_{\mathcal{M}}[T^* = t^* \mid I_i = 0]} \leq e^{\varepsilon}, \qquad e^{-2\varepsilon} \leq \frac{P_{\mathcal{M}}[T^* = t^* \mid Y_i \in \mathcal{S}, I_i = 1]}{P_{\mathcal{M}}[T^* = t^* \mid Y_i \notin \mathcal{S}, I_i = 1]} \leq e^{2\varepsilon}. \tag{10}$$

We emphasize that Lemma 1 and all subsequent results require Assumption 2. Without it, generalizable knowledge from the release can lead to arbitrarily large relative risk, as demonstrated by examples in [25, 27].

For a given function $r^*$ selected by the data holder, we can determine the $\varepsilon$ that should be used for the release. This is due to the following result relating the relative risk to $\varepsilon$.

**Theorem 1.** *Under Assumption 1 and Assumption 2, if the release of $T^* = t^*$ satisfies $\varepsilon$-DP, then*

$$r_i(p_i, q_i, t^*) \leq \frac{1}{q_i p_i + e^{-2\varepsilon}(1 - q_i)p_i + e^{-\varepsilon}(1 - p_i)}. \tag{11}$$

One can solve for $e^{-\varepsilon}$ in (11) to determine the recommended $\varepsilon$, which is given by Theorem 2.

**Theorem 2.** *For any individual $i$, fix the prior probabilities, $p_i$ and $q_i$, and a desired bound on the relative disclosure risk, $r^*(p_i, q_i)$. Define $\varepsilon_i(p_i, q_i)$ to be the function*

$$\varepsilon_i(p_i, q_i) = \begin{cases} \log\left(\frac{2p_i(1-q_i)}{\sqrt{(1-p_i)^2 + 4p_i(1-q_i)\left(\frac{1}{r^*(p_i, q_i)} - p_i q_i\right)} - (1-p_i)}\right), & \text{if } 0 < q_i < 1; \\ \log\left(\frac{1-p_i}{\frac{1}{r^*(p_i, 1)} - p_i}\right), & \text{if } q_i = 1. \end{cases} \tag{12}$$

*Under the conditions of Theorem 1, any statistic $T^* = t^*$ released under $\varepsilon$-DP with $\varepsilon \leq \varepsilon_i(p_i, q_i)$ will satisfy $r_i(p_i, q_i, t^*) \leq r^*(p_i, q_i)$.*

By Theorem 2, to achieve $r_i(p_i, q_i, t^*) \leq r^*(p_i, q_i)$ for all $(p_i, q_i)$, the agency should set $\varepsilon$ via the following minimization problem:

$$\varepsilon = \min_{p_i, q_i \in (0,1]} \varepsilon_i(p_i, q_i). \tag{13}$$

Closed forms for the $\varepsilon$ resulting from a few specific risk profiles are included in the appendix.

## 4 Using Posterior-to-prior Risks for Setting $\varepsilon$

In this section, we illustrate how an agency can use the results of Section 3 to select $\varepsilon$. Notationally, for a given quantity $x$, we use $\tilde{x}$ to indicate the agency has fixed $x$ to a particular constant.

Given $\mathcal{S}$, the agency must select a form for $r^*(p_i, q_i)$. A default choice, equivalent to the naive strategy using (7) discussed above, is to set the bound to a constant $\tilde{r} > 1$, i.e.,

$$r^*(p_i, q_i) = \tilde{r}. \tag{14}$$

As we prove in Theorem 3 in the appendix, the bound in (14) implies the agency should set $\varepsilon = \log(\tilde{r})/2$. While a constant bound on relative risk is simple, agencies that tolerate different risk profiles may be able to set $\varepsilon$ to larger values, as we now illustrate.

Consider an agency that seeks to bound the relative risks for high prior probabilities and bound the absolute disclosure risk for low prior probabilities. For example, the agency may not want adversaries whose prior probabilities are low to use $t^*$ to increase those probabilities beyond 0.10. Simultaneously, the agency may want to ensure adversaries with prior probabilities around, for example, 0.20 cannot use $t^*$ to triple their posterior probability. Such an agency can specify a risk profile that requires either the relative risk be less than some $\tilde{r} > 1$ or the absolute risk be less than some $\tilde{a} < 1$, as we now illustrate via the following two examples.

The first example is a setting inspired by a case study in [13].

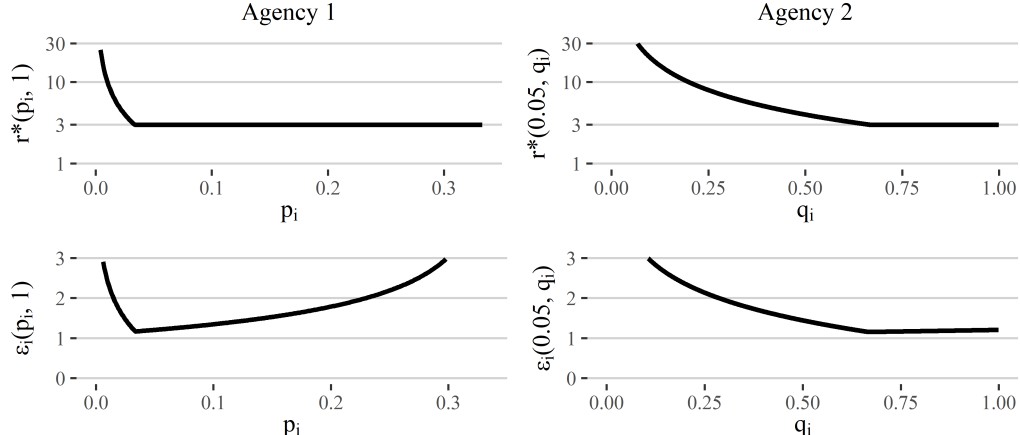

Figure 1: Each column corresponds to a particular hypothetical agency. The first row presents the agency's risk profile and the second row presents the profile's implied maximal allowable $\varepsilon$ at each point on the curve. Agency 1's risk profile is given by (15) with $\tilde{a} = 0.1$, $\tilde{q} = 1$, and $\tilde{r} = 3$, while Agency 2's risk profile is given by (16) with $\tilde{a} = 0.1$, $\tilde{p} = 0.05$, and $\tilde{r} = 3$. For Agency 2, $r^*$ and $\varepsilon_i$ are very large for small $q_i$; large values are truncated for readability.

| Agency | $\tilde{r}$ | $\tilde{a}$ | $\varepsilon$ Recommendation | Noise Std. Dev. | Prob. Exact |
|--------|------|------|------------------------------|-----------------|-------------|
| A | 1.5 | 0.25 | 0.51 | 2.74 | 25% |
| B | 3 | 0.25 | 1.30 | 1.02 | 57% |
| C | 6 | 0.25 | 2.04 | 0.59 | 77% |

Table 1: The $\varepsilon$ recommended by our framework for three risk profiles of the form in (15). Also included are the standard deviations of the noise distribution and the probabilities that the exact value is released, i.e., the mass at zero, for geometric mechanisms that satisfy the corresponding $\varepsilon$-DP.

**Example 1.** *A healthcare provider possesses a data set comprising demographic information about individuals diagnosed with a particular form of cancer in a region of interest. They plan to release the count of individuals diagnosed with this form of cancer in various demographic groups via the geometric mechanism, but are concerned this release, if insufficient noise is added, could reveal which individuals in the community have cancer. They wish to choose $\varepsilon$ appropriately.*

In this example, the primary concern is with respect to inclusion in the data set. That is, for a given individual $i$, the adversary's prior probability $p_i$ is the key quantity, whereas the $q_i$ is not as important for any given $\mathcal{S}$. The agency can fix some $\tilde{q} \in (0, 1]$, for example $\tilde{q} = 1$ to focus on adversaries who already know individual $i$'s demographic information. To simplify the analysis, we set the risk profile to $\infty$ for adversaries with $\tilde{q} \neq 1$, which indicates these adversaries are not considered. A reasonable risk profile for this agency then might be of the form, for some $\tilde{r} > 1$ and $\tilde{a} < 1$,[6]

$$r^*(p_i, q_i) = \begin{cases} \max\left\{\frac{\tilde{a}}{p_i\tilde{q}}, \tilde{r}\right\}, & \text{if } q_i = \tilde{q}; \\ \infty, & \text{if } q_i \neq \tilde{q}. \end{cases} = \begin{cases} \frac{\tilde{a}}{p_i\tilde{q}}, & \text{if } q_i = \tilde{q} \text{ and } 0 < p_i < \frac{\tilde{a}}{\tilde{q}\tilde{r}}; \\ \tilde{r}, & \text{if } q_i = \tilde{q} \text{ and } \frac{\tilde{a}}{\tilde{q}\tilde{r}} \leq p_i \leq 1; \\ \infty, & \text{if } q_i \neq \tilde{q}. \end{cases} \quad (15)$$

An example visualization of this risk function is presented in the first column of Figure 1. The upper plot displays the risk profile as a function of $p_i$ when $q_i = \tilde{q}$, and the lower plot displays the maximal $\varepsilon$ for which the relative risk bound holds for each $p_i$. We can derive a closed form for $\varepsilon$ under this risk profile; see Table 4 in Appendix A.2 for a summary and Theorem 5 in Appendix B.1 for details.

Suppose the agency is generally willing to accept a maximum absolute disclosure risk of $\tilde{a} = 0.25$ for adversaries with small prior probabilities. Table 1 presents the maximal $\varepsilon$ which satisfies the desired risk profile for three such agencies. Agency A is risk averse, agency C is utility seeking, and agency

---

[6]The second equality in (15) assumes $\tilde{q} > \tilde{a}/\tilde{r}$. If $\tilde{q} \leq \tilde{a}/\tilde{r}$, then $r^*(p_i, \tilde{q}) = \tilde{a}/(p_i\tilde{q})$ for all $0 < p_i \leq 1$.

B sits in between in terms of risk and utility. For adversaries with high prior probabilities, agencies A, B, and C bound the relative disclosure risk at $\tilde{r} = 1.5$, $\tilde{r} = 3$, and $\tilde{r} = 6$, respectively. Figure 3 in Appendix C.1 plots the forms of each agency's risk profile. To provide intuition on the amount of noise implied by these $\varepsilon$, in Table 1 we display the standard deviation of the noise distribution for each statistic under the geometric mechanism and the probability that the geometric mechanism will output the exact value of each statistic. The risk averse agency is recommended an $\varepsilon$ that results in a release with a high standard deviation and fairly low probability of releasing the exact value of the statistic. The utility seeking agency is recommended an $\varepsilon$ that results in a release with a fairly low standard deviation and high probability of releasing the exact value of the statistic.

The recommendations from these risk profiles, which are tailored to the setting and agency preferences, are higher than the recommendations from a corresponding simple risk profile of $r^*(p_i, q_i) = \tilde{r}$ for all prior probabilities. This simple profile yields $\varepsilon \approx 0.20$, $\varepsilon \approx 0.55$, and $\varepsilon \approx 0.90$ for $\tilde{r} = 1.5$, $\tilde{r} = 3$, and $\tilde{r} = 6$, respectively, less than half the corresponding $\varepsilon$ values in Table 1.

We now consider a second example, inspired by Example 16 in [45], that alters Example 1.

**Example 2.** *A survey is performed on a sample of individuals in the region of interest. 5% of the region is surveyed, and respondents are asked whether they have this particular form of cancer along with a series of demographic questions. The agency plans to release the counts of surveyed individuals who have and do not have the cancer in various demographic groups via the geometric mechanism, but is concerned this release, if insufficient noise is added, could reveal which individuals in the community have cancer. The agency wishes to choose $\varepsilon$ appropriately.*

In this example, the primary concern is with respect to the values in the data set. For ease of notation, let $Y_i$ be a $d$-vector of binary values, and let the first element, $Y_{1i}$, be an indicator for whether individual $i$ has this form of cancer. Set $\mathcal{S}$ to be the subset of the support of $Y_i$ for which individual $i$ has the cancer, i.e., $\mathcal{S} = \{y \in \{0,1\}^d : y_1 = 1\}$. For individual $i$, the adversary's $q_i$ is the key quantity, and their $p_i$ is not of interest. The agency can fix some $\tilde{p} \in (0,1]$; for example, if the agency is most concerned about adversaries whose only prior knowledge is that individual $i$ is in the population, but not whether they were surveyed, they might set $\tilde{p} = 0.05$. Alternatively, they might set $\tilde{p} = 1$ to imply an adversary that knows a priori that individual $i$ is included in $\mathbf{Y}$. A reasonable risk profile for this agency might be of the form, for some $\tilde{a} < 1$ and $\tilde{r} > 1$, [7]

$$r^*(p_i, q_i) = \begin{cases} \max\left\{\frac{\tilde{a}}{\tilde{p}q_i}, \tilde{r}\right\}, & \text{if } p_i = \tilde{p}; \\ \infty, & \text{if } p_i \neq \tilde{p}. \end{cases} = \begin{cases} \frac{\tilde{a}}{\tilde{p}q_i}, & \text{if } p_i = \tilde{p} \text{ and } 0 < q_i < \frac{\tilde{a}}{\tilde{p}\tilde{r}}; \\ \tilde{r}, & \text{if } p_i = \tilde{p} \text{ and } \frac{\tilde{a}}{\tilde{p}\tilde{r}} \leq q_i \leq 1; \\ \infty, & \text{if } p_i \neq \tilde{p}. \end{cases} \quad (16)$$

An example visualization of this risk function is presented in the second column of Figure 1. The upper plot displays the risk profile as a function of $q_i$ when $p_i = \tilde{p}$, and the lower plot displays the maximal $\varepsilon$ for which the relative risk bound holds for each $p_i$. The agency can select the smallest $\varepsilon$ on this curve for their release. We can derive a closed form the minimum; see Table 4 in Appendix A.2 for a summary and Theorem 4 in Appendix B.1 for details. Notably, it can be shown that the $\varepsilon$ selected under these two profiles is bounded below by $\log(\tilde{r})/2$ and may be much larger. We provide further exploration and visualizations for this example in Appendix C.2.

As demonstrated by these examples, the gap between the baseline strategy of (14) and our recommendation can be large. To further illustrate, suppose an agency has a simple, "point" risk profile, where $r^*(0.5, 1) = r'$ for $r' < 2$ (and $\infty$ elsewhere). The baseline recommends $\varepsilon = \log(r')/2 < 0.35$, while the recommendation from Theorem 2 can be shown to have the form $\varepsilon = \log(r'/(2 - r'))$, which diverges as $r' \to 2$. Thus, the gap between the recommendations can be arbitrarily large. While this likely does not represent a realistic disclosure risk profile of a real-world agency, it is instructive. In general, the baseline's recommendation is smaller because it is enforcing a relative risk of $r'$ for adversaries with small priors. If $r' < 2$, then an adversary with prior probability $p_i q_i = 0.001$ is restricted to a posterior probability at most 0.002, leading to a small $\varepsilon$ recommendation. If adversaries with small priors are ignored or allowed to have large relative risks, the recommendation from our method will outperform the baseline, and possibly by a substantial amount.

A particular agency's risk profile may not be characterized by one of the forms described above. For example, an agency may be equally concerned about $p_i$ and $q_i$, rather than focusing on one; see

---

[7]The second equality in (16) assumes $\tilde{p} > \tilde{a}/\tilde{r}$. If $\tilde{p} \leq \tilde{a}/\tilde{r}$, then $r^*(\tilde{p}, q_i) = \tilde{a}/(\tilde{p}q_i)$ for all $0 < q_i \leq 1$.

Appendix C.3 for an example of this setting. Or, as argued in [32], in some settings, agencies might be most concerned about the difference between the posterior and prior probabilities (rather than their ratio).[8] In such settings, it is straightforward to write down the corresponding risk function, $r^*(p_i, q_i)$, and the optimal $\varepsilon$ can be determined by numerically solving the minimization problem in (13).[9]

Regardless of the agency's desiderata for a risk profile, we recommend that they keep the following in mind when setting its functional form. First, for any region where $r^*(p_i, q_i) > 1/(p_i q_i)$, the risk profile generates a bound on the posterior probability that exceeds 1. Of course, the posterior probabilities themselves cannot exceed 1; thus, in these regions, the risk profile effectively does not bound the posterior risk. For example, an agency that sets $r^*(p_i, 1) = 3$ in the region where $p_i \geq 1/3$ (as in the left column of Figure 1) implicitly is willing to accept an unbounded $\varepsilon$ for prior probabilities $p_i \geq 1/3$. Second, when bounding the absolute disclosure risk below some $\tilde{a}$ in some region of $(p_i, q_i)$, the agency should require $p_i q_i < \tilde{a}$ in that region. When $p_i q_i = \tilde{a}$, the recommended $\varepsilon = 0$ since the data holder requires $T^*$ to offer no information about $Y_i$. This also suggests that an agency bounding absolute disclosure risk in a region of $(p_i, q_i)$ that sets $\tilde{a}$ close to some value of $p_i q_i$ in the region is willing to accept only small $\varepsilon$ values.

## 5 Managing the Trade-off in Privacy and Utility

Agencies can use the framework to manage trade-offs in disclosure risk and data utility. In particular, the agency can evaluate potential impacts of the DP algorithm using data quality metrics under different risk profiles, choosing an $\varepsilon$ that leads to a satisfactory trade-off. We demonstrate this in the below example, using data on infant mortality in Durham county, N.C. We note that these data may not require the use of DP in reality, but they do allow us to illustrate the process with public data.

**Example 3.** *Suppose that an agency in Durham county, N.C., wishes to release a differentially private version of the number of infant deaths the county experienced in the year 2020. The agency plans to use the geometric mechanism to release this value. Of particular interest is whether the county infant death rate meets the U. S. Department of Health and Human Services' Healthy People 2020 target of 6.0 or fewer deaths per 1,000 live births [21]. The agency wishes to minimize the probability that they release a noisy count that changes whether or not the 6.0 target is met; their goal is to ensure this probability is below 10%. It is public information that there were 4,012 live births in Durham county in 2020 [38].*

The primary concern in Example 3 is with respect to inclusion in the data. Thus, the agency focuses on adversaries with $q_i = 1$ and varying $p_i$. The agency is generally willing to incur large relative risks if $p_i$ is small and small relative risks if $p_i$ is large. Furthermore, due to possible policy ramifications of appearing not to meet the target rate, the agency is open to accepting greater privacy risks for greater accuracy in the released count, within reason as determined by agency decision makers.

The agency's privacy experts determine that they can characterize the agency's risk profiles reasonably well using the following general class of risk profiles.

$$r^*(p_i, q_i) = \begin{cases} \max\left\{ \frac{\tilde{a}}{p_i}, \tilde{r} \right\}, & \text{if } q_i = 1; \\ \infty, & \text{if } q_i \neq 1. \end{cases} \tag{17}$$

To allow for consideration of the effects on accuracy of different specifications of (17), the agency considers combinations of $\tilde{r} \in \{1.2, 2, 5\}$ and $\tilde{a} \in \{0.1, 0.25, 0.5\}$. They also examine $\tilde{a} = 0$, which corresponds to a constant risk profile $r^*(p_i, q_i) = \tilde{r}$. The agency could consider other combinations of parameters or risk profiles classes should these not lead to a satisfactory trade-off in risk and utility.

Figure 2 summarizes an analysis of the risk/utility trade-offs. For all $\tilde{r}$, the recommended $\varepsilon$ is larger when $\tilde{a} > 0$ than under the corresponding constant risk profile. Similar increases in $\varepsilon$ also are evident for the risk profiles of Section 4; see Appendix C. Naturally, increases in $\tilde{r}$ and $\tilde{a}$ decrease the RMSEs[10] of the noisy counts due to larger $\varepsilon$. The switching probabilities in the top panel do not use

---

[8]Requiring the difference between the posterior and prior probabilities to be bounded by some $\tilde{b} \in (0, 1)$ is equivalent to requiring the relative risk be bounded by $(p_i q_i + \tilde{b})/(p_i q_i)$. See Appendix B.2 for details.

[9]We provide R code at `https://github.com/zekicankazan/choosing_dp_epsilon` to determine the recommended $\varepsilon$ for a provided risk profile. This code was used to determine closed forms for $\varepsilon$ for the additional risk profiles described in Appendix B.2. The example in Appendix C.3 demonstrates the use of this code.

[10]Since $T^*$ is unbiased for the true count, the RMSE is the standard deviation of the noise distribution.

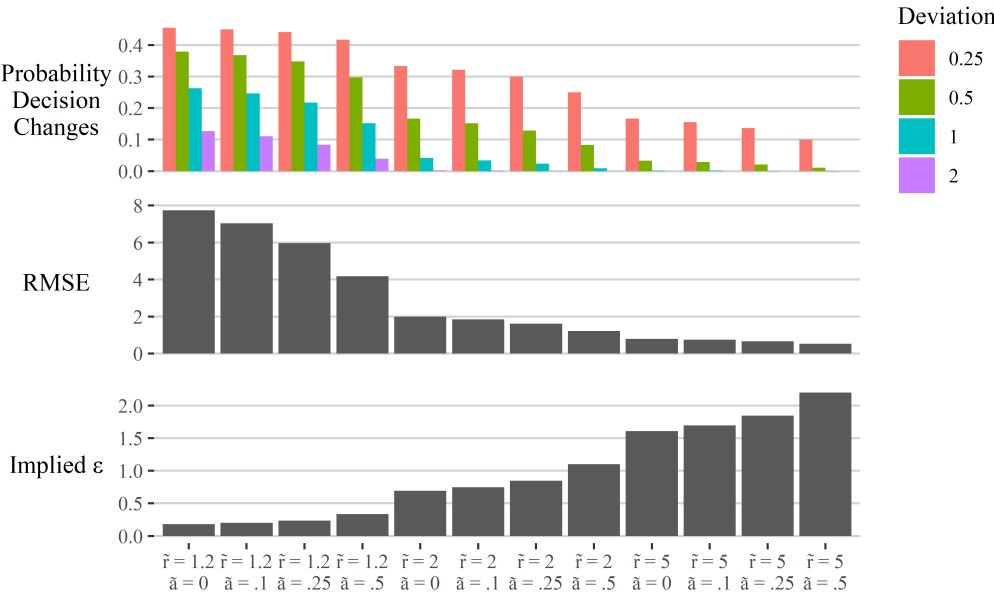

Figure 2: The top panel presents the probability that the differentially private algorithm switches whether Durham county's released rate is above or below the 6.0 target—e.g., the added noise makes the released rate 6.5 but the actual rate is 5.5. Each color represents a different hypothetical absolute difference between the true and target rate. The middle panel presents RMSEs of the noisy count of infant deaths. The bottom panel presents the implied $\varepsilon$. Each bar corresponds to a different risk profile of the form in (17).

the true count of infant deaths; rather, they consider hypothetical deviations of 0.25, 0.5, 1, and 2 in the true rate from the target rate of 6.0. In this way, the agency avoids additional privacy leakage. The probability differs substantially for each of the four deviations, with the deviation of 0.25 leading to the largest probability of a switch. To ensure the probability is below 10% for the deviation of 0.25, the agency would need to use the most aggressive of the risk profiles, which allows absolute disclosure risk of $\tilde{a} = 0.5$ or relative disclosure risk of $\tilde{r} = 5$. This leads to $\varepsilon \approx 2.20$ with RMSE $\approx 0.53$. If they are willing to allow the probability to exceed 10% or to place priority on deviations exceeding 0.25, the agency could select a less aggressive risk profile that yields a smaller $\varepsilon$.

In actuality, there were 6.2 infant deaths per 1,000 live births in Durham County in 2020 [37]. As the truth is close to the 6.0 target, the agency would need to use a large $\varepsilon$ to achieve their goal.

## 6  Relationship to Prior Work

The most similar work we are aware of is Lee and Clifton [29]'s "How Much is Enough? Choosing $\varepsilon$ for Differential Privacy," which also uses a Bayesian approach to set $\varepsilon$. The authors focus on settings where the population is public information and the adversary's goal is to determine which subset of individuals in the population was used for a differentially private release of a statistic. Their method for selecting $\varepsilon$ is tailored to the setting where only disclosure of an individual's inclusion in the data is of concern and the values of the entire population can be used to inform the choice of $\varepsilon$. [29] focuses on the setting where the Laplace mechanism is used and the population size is known; their proposal was extended to any DP mechanism by [24, 39] and settings where the population size is unknown by [34]. In Appendix D.1, we compare the recommended $\varepsilon$ from the proposal in [29] and our framework in several examples. We find that the approach of [29], for particular statistics and populations, can recommend a larger $\varepsilon$ than the one resulting from our approach. However, our framework may be used in settings where the values of the entire population are not public and settings where the privacy of the values in the release is of primary concern.

Another series of related work involves using Bayesian semantics to communicate the meaning of $\varepsilon$ to potential study participants. For an overview of recent work in this area, see [16], [36], and citations therein. We highlight an example in [45] (corrected in [46]), which considers an individual deciding whether or not to participate in a survey for which results will be released via DP with a particular $\varepsilon$. The authors suggest that the individual consider a bound on a posterior-to-posterior ratio similar to the bound in (11) to make an informed decision about whether to participate in the survey. We describe the advice of [45] using our notation in Appendix D.2. While their and our bounds have similar expressions, the goal in [45] differs from ours. They use the bound to characterize the individual's disclosure risks for a fixed $\varepsilon$, whereas we establish the agency's disclosure risk profile in order to set $\varepsilon$.

There are a number of other works that examine Bayesian semantics of $\varepsilon$-DP, which we now briefly summarize. In their seminal work proposing DP, [11] showed that, under some conditions, bounded DP is equivalent to a bound on the relative risk when all but one data point is fixed. [25] showed that bounded DP implies a bound on the total variation distance between the posterior distribution with all data points included and the posterior distribution with one data point removed. [28] showed that both bounded and unbounded DP are equivalent to a bound on the posterior-to-prior odds ratio exactly when the presence/absence/record-value of each individual is independent of that of other individuals. Finally, [27] showed that both bounded and unbounded DP imply bounds on the ratio of the posterior distribution with all data points included to the posterior with one observation replaced with a draw from the posterior with their observation removed. These works do not discuss how these semantic characterizations of DP can be used to select $\varepsilon$.

# 7 Commentary

In this article, we propose a framework for selecting $\varepsilon$ for DP by establishing disclosure risk profiles. Essentially, we provide a method for agencies to trade the problem of selecting $\varepsilon$ for a release for the problem of specifying their desired disclosure risk profile. This process involves focusing on particular classes of adversaries the agency is most concerned about—represented by the set $\mathcal{S}$—and tuning $\varepsilon$ to ensure the risk from these adversaries is sufficiently low. We emphasize that, once applied, DP will protect against all attacks with the guarantee of DP, not just the attack used to tune $\varepsilon$.

We primarily focus on the risk from one class of adversary attacks on one individual, assuming all individuals are exchangeable. If agencies assign different risk profiles for different classes of adversaries, they could repeat the analysis for each class For example, the agency might assign different risk profiles to adversaries targeting different characteristics, e.g., they may consider whether an individual has a disease more sensitive than their age. Similarly, if an agency does not treat individuals in the data as exchangeable—for example, if an agency seeks to ensure lesser disclosure risks for groups with some characteristics than for not-necessarily-disjoint groups without those characteristics —the agency could repeat this analysis for each group. In both of these settings, the agency has a decision problem on its hands. As with designing DP solutions in general, the agency must prioritize some risks over others, e.g., using decision-theoretic criteria. This is an important topic for future work and raises ethical questions regarding how assigning different risk profiles for individuals with different characteristics could affect data equity.

One avenue for future work involves incorporating a version of this framework into differentially private data analysis tools. Recently developed interfaces for DP data releases such as [24] and [35] use the framework of [29] to guide users in setting $\varepsilon$. Our framework could be similarly incorporated to provide guidance in settings that do not satisfy the assumption of [29] that the values of the population are public. Another avenue for future work involves examining how an agency can set the risk profile appropriate for a particular release. We envision agencies could do so analogously to the elicitation of utility functions in decision theory, e.g., by considering a series of bets [40]. Other future extensions involve examining whether similar results follow under weaker assumptions. This includes settings with multiple differentially private releases via existing results that relate the relative risk to DP composition theorems (e.g., S5 of [26]). This also includes extension to variants of DP—such as zero-concentrated DP [5] or approximate DP [12]—by deriving Bayesian semantics analogous to Theorem 1 or leveraging existing results [25, 27]. Additionally, it may be possible to extend the results in this article from posterior-to-prior risks to the sorts of posterior-to-posterior risks discussed in Section 6.

## Acknowledgments and Disclosure of Funding

This research was supported by NSF grant SES-2217456. We would like to thank the anonymous reviewers for their helpful feedback as well as Salil Vadhan and Jörg Drechsler for insightful discussion about a preliminary version of this work.

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

# A Supplemental Tables

## A.1 Notation Summary

Table 2 summarizes the notation of Section 2. Table 3 summarizes the definitions of key probabilistic quantities.

| Symbol | Description |
|---|---|
| $\mathbf{P}$ | Population of individuals the data is drawn from |
| $\mathbf{Y}$ | $n \times d$ confidential data set |
| $Y_i$ | Length-$d$ vector of values for individual $i$ |
| $I_i$ | Indicator for whether individual $i$ is included in $\mathbf{Y}$ |
| $\mathbf{Y}_{-i}$ | $(n-1) \times d$ matrix of the values in $\mathbf{Y}$ excluding those for individual $i$ |
| $\mathcal{S}$ | Subset of the support of $Y_i$ constituting a privacy violation |
| $r^*(p_i, q_i)$ | Function describing the agency's desired relative risk bound |
| $\mathcal{M}$ | Adversary's model for predicting $Y_i$ |
| $T^*$ | Noisy estimate of $T(\mathbf{Y})$, the function being released |

Table 2: Summary of notation.

| Symbol | Definition | Description |
|---|---|---|
| $p_i$ | $P_{\mathcal{M}}[I_i = 1]$ | Prior probability of inclusion |
| $q_i$ | $P_{\mathcal{M}}[Y_i \in \mathcal{S} \mid I_i = 1]$ | Prior probability values disclosed |
| $r_i(p_i, q_i, t^*)$ | $\frac{P_{\mathcal{M}}[Y_i \in \mathcal{S}, I_i = 1 \mid T^* = t^*]}{P_{\mathcal{M}}[Y_i \in \mathcal{S}, I_i = 1]}$ | Relative disclosure risk |
| $a_i(p_i, q_i, t^*)$ | $P_{\mathcal{M}}[Y_i \in \mathcal{S}, I_i = 1 \mid T^* = t^*]$ | Absolute disclosure risk |

Table 3: Summary of definitions.

## A.2 Closed Forms for $\varepsilon$

In this section, we provide a reference for all derived closed forms for $\varepsilon$. Corresponding formal theorem statements are provided in Appendix B.1 and discussion of closed forms not referenced in the main text is provided in Appendix B.2. Table 4 provides the closed forms.

| Risk Profile | Condition | Recommended $\varepsilon$ |
|---|---|---|
| (14); $\tilde{r}$ | | $\frac{1}{2}\log(\tilde{r})$ |
| (15) $\max\left\{\frac{\tilde{a}}{p_i\tilde{q}}, \tilde{r}\right\}$ | $0 < \tilde{q} \leq \frac{\tilde{a}}{\tilde{r}}$ | $\frac{1}{2}\log\left(\frac{\tilde{a}(1-\tilde{q})}{\tilde{q}(1-\tilde{a})}\right)$ |
| | $0 < \tilde{q} \leq \frac{1}{\tilde{r}+1}$ and $\frac{\tilde{a}}{\tilde{r}} < \tilde{q} < 1$ | $\frac{1}{2}\log\left(\frac{1-\tilde{q}}{\frac{1}{\tilde{r}}-\tilde{q}}\right)$ |
| | $\frac{1}{\tilde{r}+1} < \tilde{q} < 1$ and $\frac{\tilde{a}}{\tilde{r}} < \tilde{q} < 1$ | $\log\left(\frac{2\tilde{a}(1-\tilde{q})}{\sqrt{(\tilde{r}\tilde{q}-\tilde{a})^2+4\tilde{a}\tilde{q}(1-\tilde{q})(1-\tilde{a})}-(\tilde{r}\tilde{q}-\tilde{a})}\right)$ |
| | $\tilde{q} = 1$ | $\log\left(\frac{\tilde{r}-\tilde{a}}{1-\tilde{a}}\right)$ |
| (16) $\max\left\{\frac{\tilde{a}}{\tilde{p}q_i}, \tilde{r}\right\}$ | $0 < \tilde{p} \leq \frac{\tilde{a}}{\tilde{r}}$ | $\log\left(\frac{\tilde{a}(1-\tilde{p})}{\tilde{p}(1-\tilde{a})}\right)$ |
| | $\frac{\tilde{a}}{\tilde{r}} < \tilde{p} \leq 1$ | $\log\left(\frac{2(\tilde{p}\tilde{r}-\tilde{a})}{\sqrt{\tilde{r}^2(1-\tilde{p})^2+4(\tilde{p}\tilde{r}-\tilde{a})(1-\tilde{a})}-\tilde{r}(1-\tilde{p})}\right)$ |
| (25) $\tilde{r}$ if $\tilde{p}_0 \leq p_i \leq \tilde{p}_1$ and $\tilde{q}_0 \leq q_i \leq \tilde{q}_1$ | $0 \leq \tilde{q}_0 \leq \frac{1}{\tilde{r}+1}$ | $\log\left(\frac{2\tilde{p}_1(1-\tilde{q}_0)}{\sqrt{(1-\tilde{p}_1)^2+4\tilde{p}_1(1-\tilde{q}_0)\left(\frac{1}{\tilde{r}}-\tilde{p}_1\tilde{q}_0\right)}-(1-\tilde{p}_1)}\right)$ |
| | $\frac{1}{\tilde{r}+1} < \tilde{q}_0 < 1$ and $\tilde{p}_0 > 0$ | $\log\left(\frac{2\tilde{p}_0(1-\tilde{q}_0)}{\sqrt{(1-\tilde{p}_0)^2+4\tilde{p}_0(1-\tilde{q}_0)\left(\frac{1}{\tilde{r}}-\tilde{p}_0\tilde{q}_0\right)}-(1-\tilde{p}_0)}\right)$ |
| | $\frac{1}{\tilde{r}+1} < \tilde{q}_0 < 1$ and $\tilde{p}_0 = 0$ | $\log(\tilde{r})$ |
| | $\tilde{q}_0 = 1$ | $\log\left(\frac{1-\tilde{p}_0}{\frac{1}{\tilde{r}}-\tilde{p}_0}\right)$ |
| (28); $\frac{p_iq_i+\tilde{b}}{p_iq_i}$ | | $\log\left(\frac{1+\tilde{b}}{1-\tilde{b}}\right)$ |

Table 4: Closed forms for $\varepsilon$ for various $r^*(p_i, q_i)$.

# B  Additional Results

## B.1  Omitted Results

In this section, we present all results omitted from the paper. Proofs of these results are included in Appendix E.

First, we state a corollary to Theorem 1, which generalizes Theorem 1.3 in [19] to sets where $|\mathcal{S}| > 1$ and to unbounded DP.

**Corollary 1.** *Under the conditions of Theorem 1, for all $p_i, q_i \in (0, 1]$ and all $t^*$,*

$$r_i(p_i, q_i, t^*) \le e^{2\varepsilon}. \tag{18}$$

Next, we state a corollary to Theorem 2 which considers the special case where $p_i = 1$. This corresponds to a setting where individual $i$'s inclusion in the data is known a priori by the adversary, for example data from a census or public social media platform.

**Corollary 2.** *Under the conditions of Theorem 2, if $p_i = 1$ and $0 < q_i < 1$, any statistic $T^* = t^*$ released under $\varepsilon$-DP with*

$$\varepsilon \le \frac{1}{2} \log \left( \frac{1 - q_i}{\frac{1}{r^*(1, q_i)} - q_i} \right), \tag{19}$$

*will satisfy $r_i(1, q_i, t^*) \le r^*(1, q_i)$.*

For particular forms of $r^*$, the optimization in (13) has a closed form solution. This is detailed by the following theorems, which are preceded by two lemmas used in the proofs.

**Lemma 2.** *Fix $p_i, q_i \in (0, 1)$ and let $r^*(p_i, q_i) = \frac{\tilde{a}}{p_i q_i}$. Then the function*

$$\varepsilon(p_i, q_i) = \log \left( \frac{2 p_i (1 - q_i)}{\sqrt{(1 - p_i)^2 + 4 p_i (1 - q_i) \left( \frac{1}{r^*(p_i, q_i)} - p_i q_i \right)} - (1 - p_i)} \right) \tag{20}$$

*has partial derivatives such that*

1. *$\frac{\partial \varepsilon(p_i, q_i)}{\partial p_i} < 0$ for all $0 < p_i < 1$ and $0 < q_i < 1$*

2. *$\frac{\partial \varepsilon(p_i, q_i)}{\partial q_i} < 0$ for all $0 < p_i < 1$ and $0 < q_i < 1$.*

**Lemma 3.** *Fix $p_i, q_i \in (0, 1)$ and let $r^*(p_i, q_i) = \tilde{r}$. Then the function*

$$\varepsilon(p_i, q_i) = \log \left( \frac{2 p_i (1 - q_i)}{\sqrt{(1 - p_i)^2 + 4 p_i (1 - q_i) \left( \frac{1}{r^*(p_i, q_i)} - p_i q_i \right)} - (1 - p_i)} \right) \tag{21}$$

*has partial derivatives such that*

1. *$\frac{\partial \varepsilon(p_i, q_i)}{\partial p_i} < 0$ if $0 < q_i < \frac{1}{\tilde{r}+1}$ and $0 < p_i < 1$*

2. *$\frac{\partial \varepsilon(p_i, q_i)}{\partial p_i} = 0$ if $q_i = \frac{1}{\tilde{r}+1}$ and $0 < p_i < 1$*

3. *$\frac{\partial \varepsilon(p_i, q_i)}{\partial p_i} > 0$ if $\frac{1}{\tilde{r}+1} < q_i < 1$ and $0 < p_i < 1$*

4. *$\frac{\partial \varepsilon(p_i, q_i)}{\partial q_i} > 0$ for all $0 < p_i < 1$ and $0 < q_i < 1$.*

**Theorem 3.** *Under the conditions of Theorem 2, if $r^*(p_i, q_i) = \tilde{r} > 1$, the solution to the minimization problem in (13) is*

$$\varepsilon = \frac{1}{2} \log (\tilde{r}). \tag{22}$$

**Theorem 4.** *Under the conditions of Theorem 2, let $\tilde{a} < 1$, $\tilde{p} \le 1$, and $\tilde{r} > 1$, and $0 < q_i < 1$. If the function $r^*$ is such that $r^*(\tilde{p}, q_i) = \max\{\tilde{a}/(\tilde{p}q_i), \tilde{r}\}$ and $r^*(p_i, q_i) = \infty$ if $p_i \ne \tilde{p}$, then the solution to the minimization problem in (13) is*

$$
\varepsilon = \begin{cases} \log\left(\frac{\tilde{a}(1-\tilde{p})}{\tilde{p}(1-\tilde{a})}\right), & \text{if } 0 < \tilde{p} \le \frac{\tilde{a}}{\tilde{r}}; \\ \log\left(\frac{2(\tilde{p}\tilde{r}-\tilde{a})}{\sqrt{\tilde{r}^2(1-\tilde{p})^2+4(\tilde{p}\tilde{r}-\tilde{a})(1-\tilde{a})}-\tilde{r}(1-\tilde{p})}\right), & \text{if } \frac{\tilde{a}}{\tilde{r}} < \tilde{p} \le 1. \end{cases}
\tag{23}
$$

**Theorem 5.** *Under the conditions of Theorem 2, let $\tilde{a} < 1$, $\tilde{q} \le 1$, and $\tilde{r} > 1$, and $0 < p_i < 1$. If the function $r^*$ is such that $r^*(p_i, \tilde{q}) = \max\{\tilde{a}/(p_i\tilde{q}), \tilde{r}\}$ and $r^*(p_i, q_i) = \infty$ for $q_i \ne \tilde{q}$, then the solution to the minimization problem in (13) is*

$$
\varepsilon = \begin{cases} \frac{1}{2}\log\left(\frac{\tilde{a}(1-\tilde{q})}{\tilde{q}(1-\tilde{a})}\right), & \text{if } 0 < \tilde{q} \le \frac{\tilde{a}}{\tilde{r}}; \\ \frac{1}{2}\log\left(\frac{1-\tilde{q}}{\frac{1}{\tilde{r}}-\tilde{q}}\right), & \text{if } 0 < \tilde{q} \le \frac{1}{\tilde{r}+1} \text{ and } \frac{\tilde{a}}{\tilde{r}} < \tilde{q} < 1; \\ \log\left(\frac{2\tilde{a}(1-\tilde{q})}{\sqrt{(\tilde{r}\tilde{q}-\tilde{a})^2+4\tilde{a}\tilde{q}(1-\tilde{q})(1-\tilde{a})}-(\tilde{r}\tilde{q}-\tilde{a})}\right), & \text{if } \frac{1}{\tilde{r}+1} < \tilde{q} < 1 \text{ and } \frac{\tilde{a}}{\tilde{r}} < \tilde{q} < 1; \\ \log\left(\frac{\tilde{r}-\tilde{a}}{1-\tilde{a}}\right), & \text{if } \tilde{q} = 1. \end{cases}
\tag{24}
$$

## B.2 Additional Closed Forms

In this section, we present closed form results for the $\varepsilon$ implied by additional disclosure risk profiles not discussed in the main text. Proofs of these results are not provided, although we note that proofs would have a structure similar to the proofs of Theorems 3–5. It is straightforward to verify these results empirically.

First, consider an agency that requires a constant relative risk bound to hold on a subset of the $(p_i, q_i)$ space and does not have any bounds outside that space. That is, rather than enforcing $r^*(p_i, q_i) = \tilde{r}$ for all $0 \le p_i, q_i \le 1$, the agency only enforces this condition for $\tilde{p}_0 \le p_i \le \tilde{p}_1$ and $\tilde{q}_0 \le q_i \le \tilde{q}_1$, where $0 \le \tilde{p}_0 \le \tilde{p}_1 \le 1$ and $0 \le \tilde{q}_0 \le \tilde{q}_1 \le 1$ (for $\tilde{p}_1, \tilde{q}_1 > 0$). Formally,

$$
r^*(p_i, q_i) = \begin{cases} \tilde{r}, & \text{if } \tilde{p}_0 \le p_i \le \tilde{p}_1 \text{ and } \tilde{q}_0 \le q_i \le \tilde{q}_1; \\ \infty, & \text{otherwise.} \end{cases}
\tag{25}
$$

The recommended $\varepsilon$ for an agency with the risk profile in (25) is

$$
\varepsilon = \begin{cases} \log\left(\frac{2\tilde{p}_1(1-\tilde{q}_0)}{\sqrt{(1-\tilde{p}_1)^2+4\tilde{p}_1(1-\tilde{q}_0)\left(\frac{1}{\tilde{r}}-\tilde{p}_1\tilde{q}_0\right)}-(1-\tilde{p}_1)}\right), & \text{if } 0 \le \tilde{q}_0 \le \frac{1}{\tilde{r}+1}; \\ \log\left(\frac{2\tilde{p}_0(1-\tilde{q}_0)}{\sqrt{(1-\tilde{p}_0)^2+4\tilde{p}_0(1-\tilde{q}_0)\left(\frac{1}{\tilde{r}}-\tilde{p}_0\tilde{q}_0\right)}-(1-\tilde{p}_0)}\right), & \text{if } \frac{1}{\tilde{r}+1} < \tilde{q}_0 < 1 \text{ and } \tilde{p}_0 > 0; \\ \log(\tilde{r}), & \text{if } \frac{1}{\tilde{r}+1} < \tilde{q}_0 < 1 \text{ and } \tilde{p}_0 = 0; \\ \log\left(\frac{1-\tilde{p}_0}{\frac{1}{\tilde{r}}-\tilde{p}_0}\right), & \text{if } \tilde{q}_0 = 1. \end{cases}
\tag{26}
$$

Next, consider an agency that desires a constant bound on the difference between the posterior and the prior. That is, for some $\tilde{b} \in (0, 1)$, they would like to enforce

$$
P_{\mathcal{M}}[Y_i \in \mathcal{S}, I_i = 1 \mid T^* = t^*] - P_{\mathcal{M}}[Y_i \in \mathcal{S}, I_i = 1] \le \tilde{b}.
\tag{27}
$$

This constraint corresponds to the following risk profile.

$$
r^*(p_i, q_i) = \frac{p_i q_i + \tilde{b}}{p_i q_i} = 1 + \frac{\tilde{b}}{p_i q_i}.
\tag{28}
$$

The recommended $\varepsilon$ for an agency with this risk profile is

$$
\varepsilon = \log\left(\frac{1+\tilde{b}}{1-\tilde{b}}\right).
\tag{29}
$$

Notably, by a Taylor series approximation, $-\log(1-\tilde{b}) \approx \tilde{b} + \tilde{b}^2/2$. Thus, for small $\tilde{b}$, it follows that the recommended $\varepsilon$ can be approximated by

$$
\varepsilon = \log(1+\tilde{b}) - \log(1-\tilde{b}) \approx \left(\tilde{b} - \frac{\tilde{b}^2}{2}\right) + \left(\tilde{b} + \frac{\tilde{b}^2}{2}\right) = 2\tilde{b}.
\tag{30}
$$

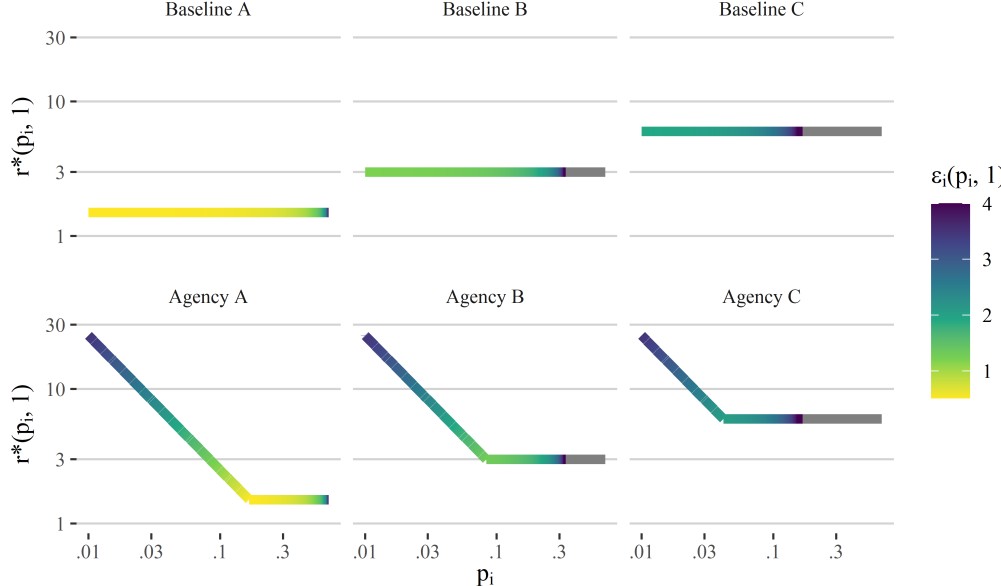

Figure 3: The risk profiles for three agencies with risk profile given by (31). The lines in the lower panels represent the risk profiles for $q_i = 1$ as a function of $p_i$, and the colors represent the implied $\varepsilon$ at each point on the curve. The lines in the upper panels represent the corresponding baseline $r^*(p_i, 1) = \tilde{r}$. The left plots set $\tilde{r} = 1.5$, the center plots set $\tilde{r} = 3$, and the right plots set $\tilde{r} = 6$.

## C  Additional Examples

This section provides additional examples of risk profiles.

### C.1  Additional Figure for Example 1

Here we provide an additional figure for Example 1. The setting is restated below.

**Example 1.** *A healthcare provider possesses a data set comprising demographic information about individuals diagnosed with a particular form of cancer in a region of interest. They plan to release the count of individuals diagnosed with this form of cancer in various demographic groups via the geometric mechanism, but are concerned this release, if insufficient noise is added, could reveal which individuals in the community have cancer. They wish to choose $\varepsilon$ appropriately.*

As discussed in Section 4, a reasonable risk profile for this setting might focus on $p_i$ and set $q_i = 1$. If we additionally suppose the agency is generally willing to accept a maximum absolute disclosure risk of $0.25$ for adversaries with small prior probabilities. A risk profile for this agency might be of the form, for some $\tilde{r} > 1$,

$$r^*(p_i, q_i) = \begin{cases} \max\left\{\frac{0.25}{p_i}, \tilde{r}\right\}, & \text{if } q_i = 1; \\ \infty, & \text{if } q_i \neq 1. \end{cases} \tag{31}$$

The three example risk functions considered in Section 4 are presented in Figure 3. Agency A is risk averse, agency C is utility seeking, and agency B sits in between in terms of risk and utility. For adversaries with high prior probabilities, agencies A, B, and C bound the relative disclosure risk at $\tilde{r} = 1.5$, $\tilde{r} = 3$, and $\tilde{r} = 6$, respectively.

### C.2  Extended Example 2

Here we provide an extended analysis of Example 2. The setting is restated below.

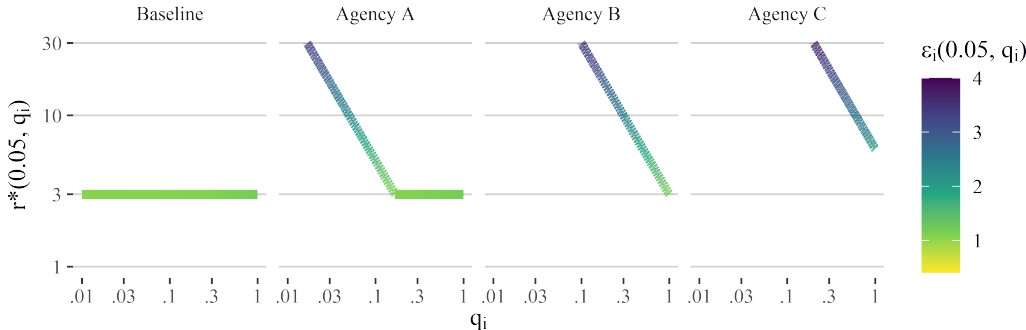

Figure 4: The risk profiles for three agencies with risk profile given by (32). The lines represent the risk profiles for $p_i = 0.05$ as a function of $q_i$, and the colors represent the implied $\varepsilon$ at each point on the curve. The left plot sets $\tilde{a} = 0.025$, the center sets $\tilde{a} = 0.15$, and the right sets $\tilde{a} = 0.3$.

| Agency | $\tilde{a}$ | $\varepsilon$ Recommendation | Noise Std. Dev. | Prob. Exact |
|--------|------|------|------|------|
| A | 0.025 | 1.09 | 1.24 | 50% |
| B | 0.15 | 1.21 | 1.10 | 54% |
| C | 0.3 | 2.10 | 0.56 | 78% |

Table 5: For each of the three risk profiles in Figure 4, we present the $\varepsilon$ recommended by our framework. For a release satisfying $\varepsilon$-DP using the geometric mechanism, we present the corresponding standard deviation of the noise distribution and the probability that the exact value is released (i.e., the noise distribution's probability mass at zero).

**Example 2.** *A survey is performed on a sample of individuals in the region of interest. 5% of the region is surveyed, and respondents are asked whether they have this particular form of cancer along with a series of demographic questions. The agency plans to release the counts of surveyed individuals who have and do not have the cancer in various demographic groups via the geometric mechanism, but is concerned this release, if insufficient noise is added, could reveal which individuals in the community have cancer. The agency wishes to choose $\varepsilon$ appropriately.*

As discussed in Section 4, a reasonable risk profile for this setting might set $\mathcal{S} = \{y \in \{0, 1\}^d : y_1 = 1\}$ and focus on $q_i$, while fixing $p_i = 0.05$. Additionally, suppose the agency is generally willing to accept a maximum relative disclosure risk of 3 for adversaries with large prior probabilities. A reasonable risk profile for this agency might be of the form, for some $\tilde{a} < 1$,

$$r^*(p_i, q_i) = \begin{cases} \max\left\{\frac{\tilde{a}}{0.05q_i}, 3\right\}, & \text{if } p_i = 0.05; \\ \infty, & \text{if } p_i \neq 0.05. \end{cases} \tag{32}$$

Three example risk functions of this form are presented in Figure 4. Once again, agency A is risk averse, agency C is utility seeking, and agency B sits in between on risk and utility. Agencies A, B, and C are willing to allow adversaries to achieve an absolute disclosure risk of $\tilde{a} = 0.025$, $\tilde{a} = 0.15$, and $\tilde{a} = 0.3$, respectively. Table 5 presents the $\varepsilon$ recommendations for each agency along with the standard deviation of the noise and probability of releasing the exact value of each statistic under the geometric mechanism.

As in Table 1, the $\varepsilon$ recommendations reflect trade offs between privacy and accuracy. They also are much higher than the recommendations from a corresponding simple risk profile of $r^*(p_i, q_i) = 3$ for all prior probabilities, which implies $\varepsilon \approx 0.55$. Even the most risk averse agency is recommended an $\varepsilon$ that is much larger than this baseline risk profile. This gain is primarily due to the assumptions that the survey is a simple random sample from the population, and the adversary has no prior knowledge about which individuals are surveyed. Essentially, the additional uncertainty from the sampling mechanism allows for an $\varepsilon$ recommendation with less noise injected. This is consistent with prior work showing that DP mechanisms applied to random subsamples provide better privacy guarantees [3]. The recommended $\varepsilon$ will increase as the proportion of individuals from the population surveyed

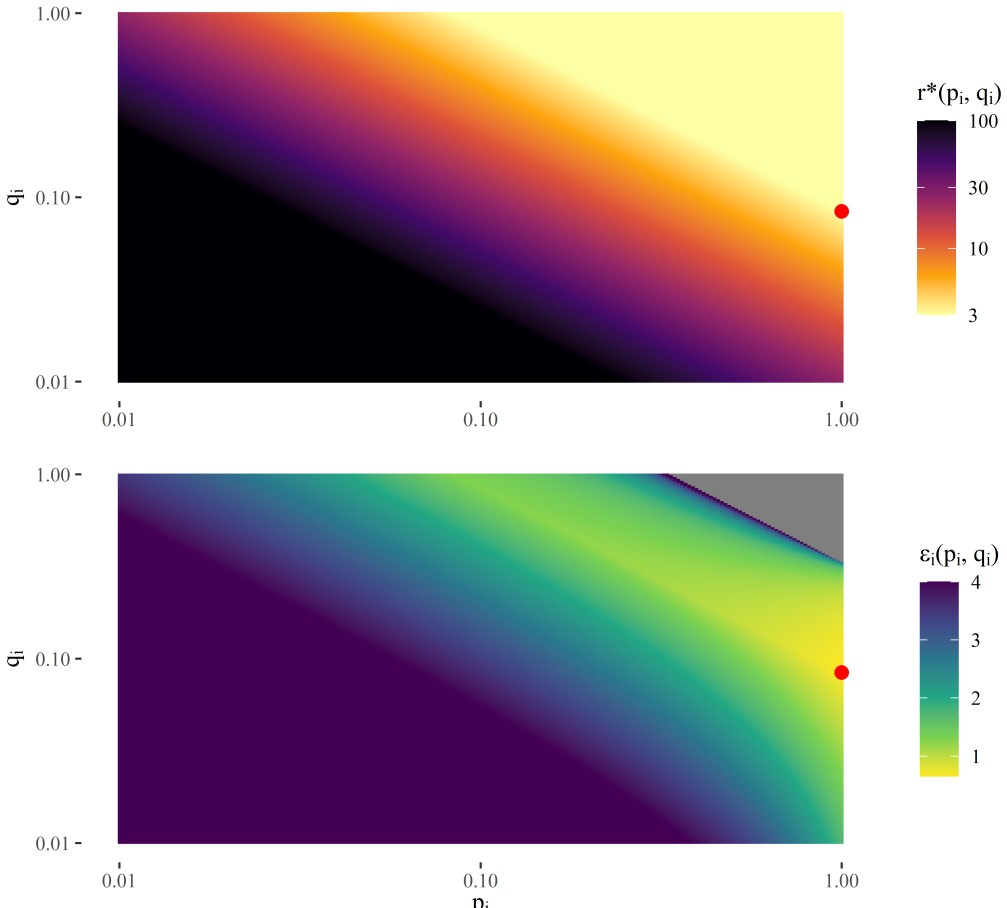

Figure 5: The top panel presents the $r^*(p_i, q_i)$ from (33) as a function of $p_i$ and $q_i$. The bottom panel presents the implied $\varepsilon_i(p_i, q_i)$ as a function of $p_i$ and $q_i$. The red point represents $\mathrm{argmin}_{(p_i, q_i)} \, \varepsilon_i(p_i, q_i)$. For clarity of presentation, all $r^*(p_i, q_i) > 100$ are truncated to 100 and all $\varepsilon_i(p_i, q_i) > 4$ are truncated to 4.

($\tilde{p}$) decreases. For example, for the risk function in (32) with $\tilde{a} = 0.025$ and $\tilde{r} = 3$, if $\tilde{p} = 0.005$, we recommend $\varepsilon \approx 1.63$; if $\tilde{p} = 0.0005$, we recommend $\varepsilon \approx 3.94$.

### C.3 Two-dimensional Example

The examples in Appendices C.1 and C.2 focus on settings where only one of $p_i$ and $q_i$ is important and the other can be reasonably fixed to a constant. Now we consider the setting where both may be simultaneously important. For example, in Example 2, the agency may wish to consider adversaries with various $p_i$, rather than only $\tilde{p} = 0.05$. Perhaps the agency wishes to limit the absolute disclosure to be below $\tilde{a} = 0.25$ when total prior probability of disclosure, $p_i q_i$, is low and limit relative disclosure to be below $\tilde{r} = 3$ when $p_i q_i$ is high. This corresponds to the following risk profile.

$$r^*(p_i, q_i) = \max \left\{ \frac{\tilde{a}}{p_i q_i}, \tilde{r} \right\} = \max \left\{ \frac{0.25}{p_i q_i}, 3 \right\}. \tag{33}$$

A plot of this desired bound on pairs $(p_i, q_i)$ on the space $(0, 1] \times (0, 1]$ is presented in the top panel of Figure 5. Notably, if $p_i$ is fixed at $\tilde{p} \in (0, 1]$, a plot of $r^*(\tilde{p}, q_i)$ as a function of $q_i$ has a form similar to the plots of Figure 4. Similarly, if $q_i$ is fixed at $\tilde{q} \in (0, 1]$, a plot of $r^*(p_i, \tilde{q})$ as a function of $p_i$ has a form similar to the plots of Figure 3.

To determine the $\varepsilon$ recommended by this profile, we numerically solve the minimization problem in (13). A plot of $\varepsilon_i(p_i, q_i)$ as a function of pairs $(p_i, q_i)$ on the space $(0, 1] \times (0, 1]$ is presented in the bottom panel of Figure 5. Our framework recommends $\varepsilon \approx 0.65$, which corresponds to $p_i = 1$ and $q_i = 0.083 = \tilde{a}/\tilde{r}$.

# D   Additional Discussion of Prior Work

In this section, we describe two previous works with similar aims to our framework using the notation in Section 2.2, expanding on the discussion in Section 6.

## D.1   "How Much is Enough? Choosing $\varepsilon$ for Differential Privacy"

[29] focus on settings where the population, $\mathbf{P}$, of size $n$ is public information and the adversary's goal is to determine which subset of individuals in $\mathbf{P}$ was used for a differentially private release of a statistic. We can characterize their setting with the notation of Section 2.2 as follows. We define $\mathbf{Y}$ to be the subset of individuals' values in $\mathbf{P}$ used to compute the statistic of interest, $T(\mathbf{Y})$, and its released DP counterpart, $T^*(\mathbf{Y})$. In their examples, the authors focus on the setting where only one individual is removed from $\mathbf{P}$ to create $\mathbf{Y}$, and the adversary's goal is to determine which $i$ was removed.

We can apply our framework to this setting with a minor modification. For this comparison, we assume the adversary's $q_i = P_{\mathcal{M}}[Y_i \in \mathcal{S} \mid I_i = 0] = 1$ for any set $\mathcal{S}$ (although we note that this is a weaker assumption than that of Lee and Clifton, since we do not assume $\mathbf{P}$ is public). We redefine $p_i$ and the risk measures to be in terms of $I_i = 0$, rather than $I_i = 1$.

$$p_i = P_{\mathcal{M}}[I_i = 0] \tag{34}$$

$$r_i(p_i, 1, t^*) = \frac{P_{\mathcal{M}}[I_i = 0 \mid T^* = t^*]}{P_{\mathcal{M}}[I_i = 0]} \tag{35}$$

$$a_i(p_i, 1, t^*) = P_{\mathcal{M}}[I_i = 0 \mid T^* = t^*]. \tag{36}$$

[29] focus on the case of $p_i = 1/n$, and seek to enforce the bound $a_i(1/n, 1, t^*) \leq \tilde{a}$ for some constant $\tilde{a}$ and all $t^*$, which implies the relative risk bound

$$r^*(p_i, q_i) = \begin{cases} n\tilde{a}, & \text{if } p_i = \frac{1}{n}, q_i = 1; \\ \infty, & \text{otherwise.} \end{cases} \tag{37}$$

From an analogy to Theorem 2 with the redefined $p_i$, it follows that under these conditions, our method sets

$$\varepsilon = \log\left(\frac{1 - \frac{1}{n}}{\frac{1}{n\tilde{a}} - \frac{1}{n}}\right) = \log\left(\frac{(n-1)\tilde{a}}{1 - \tilde{a}}\right). \tag{38}$$

In the motivating example from their paper, the authors set $n = 4$ and $\tilde{a} = 1/3$, giving $r^*(1/n, 1) = 4/3$. This results in $\varepsilon = \log(3/2) \approx 0.41$ from our method. When the release mechanism is the addition of Laplace noise, [29]'s method arrives at a similar form, but with the recommendation scaled by a factor of $\Delta T/\Delta v$.

$$\varepsilon = \frac{\Delta T}{\Delta v} \log\left(\frac{(n-1)\tilde{a}}{1 - \tilde{a}}\right), \tag{39}$$

$$\Delta T = \max\left\{|T(\mathbf{Y}_1) - T(\mathbf{Y}_2)| : \mathbf{Y}_2 \subset \mathbf{Y}_1 \subset \mathbf{P}, |\mathbf{Y}_1| = n - 1, |\mathbf{Y}_2| = n - 2\right\} \tag{40}$$

$$\Delta v = \max\left\{|T(\mathbf{Y}_1) - T(\mathbf{Y}_2)| : \mathbf{Y}_1 \subset \mathbf{P}, \mathbf{Y}_2 \subset \mathbf{P}, |\mathbf{Y}_1| = |\mathbf{Y}_2| = n - 1\right\}. \tag{41}$$

The recommendation of [29]'s method thus depends on both the population and the particular release function. This results in different $\varepsilon$ values for the same $n$ and $\tilde{a}$, as low as 0.34 and as high as 1.62 in the authors' examples, depending on the statistic of interest and the values in the data.

## D.2   "Differential Privacy: A Primer for a Non-Technical Audience"

The example in [45] considers an individual deciding whether or not to participate in a survey for which results will be released via DP with a particular $\varepsilon$. Using our notation, let $Z_i = f(Y_i, I_i) \in$

$\{0, 1\}$ be a quantity of interest to the adversary, who wishes to learn whether $Z_i = 1$. They have some prior $q_i = P_{\mathcal{M}}[Z_i = 1]$. Rather than considering the relative or absolute disclosure risk, the individual is interested in comparing the adversary's posterior probability if they participate in the survey, $a_{1i}(q_i, t^*) = P_{\mathcal{M}}[Z_i = 1 \mid I_i = 1, T^* = t^*]$, to the adversary's posterior probability if they do not participate, $a_{0i}(q_i, t^*) = P_{\mathcal{M}}[Z_i = 1 \mid I_i = 0, T^* = t^*]$. The authors of [45] state that for all $q_i$ and all $t^*$,

$$a_{1i}(q_i, t^*) \leq \frac{a_{0i}(q_i, t^*)}{a_{0i}(q_i, t^*) + e^{-2\varepsilon}(1 - a_{0i}(q_i, t^*))}. \tag{42}$$

This expression is in the same spirit as the results from our framework with $p_i = 1$. By Theorem 1, we have

$$r_i(1, q_i, t^*) \leq \frac{1}{q_i + e^{-2\varepsilon}(1 - q_i)} \quad \implies \quad a_i(1, q_i, t^*) \leq \frac{q_i}{q_i + e^{-2\varepsilon}(1 - q_i)}. \tag{43}$$

The authors of [45] suggest that the individual considering survey participation use (42) to bound $a_{1i}$ for various values of $a_{0i}$. The individual can examine these bounds to make an informed decision about whether to participate in the survey.

# E  Proofs

This section provides proofs of results from Section 3 and Appendix B.1.

**Lemma 1.** *Under Assumption 1 and Assumption 2, if the release of $T^* = t^*$ satisfies DP, then for any subset $\mathcal{S}$ of the domain of $Y_i$, we have*

$$e^{-\varepsilon} \leq \frac{P_{\mathcal{M}}[T^* = t^* \mid Y_i \in \mathcal{S}, I_i = 1]}{P_{\mathcal{M}}[T^* = t^* \mid I_i = 0]} \leq e^{\varepsilon}, \qquad e^{-2\varepsilon} \leq \frac{P_{\mathcal{M}}[T^* = t^* \mid Y_i \in \mathcal{S}, I_i = 1]}{P_{\mathcal{M}}[T^* = t^* \mid Y_i \notin \mathcal{S}, I_i = 1]} \leq e^{2\varepsilon}. \tag{10}$$

*Proof.* To begin, we demonstrate that for all $y$ in the support of $Y_i$ and $t^*$ in the support of $T^*$,

$$e^{-\varepsilon} \leq \frac{P_{\mathcal{M}}[T^* = t^* \mid Y_i = y, I_i = 1]}{P_{\mathcal{M}}[T^* = t^* \mid I_i = 0]} \leq e^{\varepsilon}. \tag{44}$$

Let $\mathcal{Y}_{-i}$ be the support of $\mathbf{Y}_{-i}$ under $\mathcal{M}$. Then,

$$P_{\mathcal{M}}[T^* = t^* \mid Y_i = y, I_i = 1]$$

$$= \sum_{\mathbf{y}_{-i} \in \mathcal{Y}_{-i}} P[T^* = t^* \mid Y_i = y, \mathbf{Y}_{-i} = \mathbf{y}_{-i}, I_i = 1] \, P_{\mathcal{M}}[\mathbf{Y}_{-i} = \mathbf{y}_{-i} \mid Y_i = y, I_i = 1] \tag{45}$$

$$\leq \sum_{\mathbf{y}_{-i} \in \mathcal{Y}_{-i}} e^{\varepsilon} P[T^* = t^* \mid \mathbf{Y}_{-i} = \mathbf{y}_{-i}, I_i = 0] \, P_{\mathcal{M}}[\mathbf{Y}_{-i} = \mathbf{y}_{-i} \mid Y_i = y, I_i = 1] \tag{46}$$

$$= e^{\varepsilon} \sum_{\mathbf{y}_{-i} \in \mathcal{Y}_{-i}} P_{\mathcal{M}}[T^* = t^* \mid \mathbf{Y}_{-i} = \mathbf{y}_{-i}, I_i = 0] \, P[\mathbf{Y}_{-i} = \mathbf{y}_{-i} \mid I_i = 0] \tag{47}$$

$$= e^{\varepsilon} P_{\mathcal{M}}[T^* = t^* \mid I_i = 0]. \tag{48}$$

The equality in (45) follows from the law of total probability and Assumption 1. The inequality in (46) follows from DP via (1). The equality in (47) follows from Assumption 2. The equality in (48) follows from the law of total probability and Assumption 1. This completes the proof of the right inequality. The proof of the left inequality is identical with the other inequality in (1) applied in (46).

Now, we apply Bayes' Theorem to $P_{\mathcal{M}}[T^* = t^* \mid Y_i \in \mathcal{S}, I_i = 1]$.

$$\frac{P_{\mathcal{M}}[T^* = t^* \mid Y_i \in \mathcal{S}, I_i = 1]}{P_{\mathcal{M}}[T^* = t^* \mid I_i = 0]} = \frac{\frac{P_{\mathcal{M}}[Y_i \in \mathcal{S} \mid T^* = t^*, I_i = 1] \, P_{\mathcal{M}}[T^* = t^* \mid I_i = 1]}{P_{\mathcal{M}}[Y_i \in \mathcal{S} \mid I_i = 1]}}{P_{\mathcal{M}}[T^* = t^* \mid I_i = 0]} \tag{49}$$

$$= \frac{P_{\mathcal{M}}[T^* = t^* \mid I_i = 1] \, P_{\mathcal{M}}[Y_i \in \mathcal{S} \mid T^* = t^*, I_i = 1]}{P_{\mathcal{M}}[Y_i \in \mathcal{S} \mid I_i = 1] \, P_{\mathcal{M}}[T^* = t^* \mid I_i = 0]} \tag{50}$$

We can then break the second term in the numerator into a summation and apply Bayes' Theorem to each term in the sum.

$$\frac{P_{\mathcal{M}}[T^* = t^* \mid Y_i \in \mathcal{S}, I_i = 1]}{P_{\mathcal{M}}[T^* = t^* \mid I_i = 0]}$$

$$= \frac{P_{\mathcal{M}}[T^* = t^* \mid I_i = 1] \sum_{y \in \mathcal{S}} P_{\mathcal{M}}[Y_i = y \mid T^* = t^*, I_i = 1]}{P_{\mathcal{M}}[Y_i \in \mathcal{S} \mid I_i = 1] \, P_{\mathcal{M}}[T^* = t^* \mid I_i = 0]} \tag{51}$$

$$= \frac{P_{\mathcal{M}}[T^* = t^* \mid I_i = 1] \sum_{y \in \mathcal{S}} \frac{P_{\mathcal{M}}[T^* = t^* \mid Y_i = y, I_i = 1] \, P_{\mathcal{M}}[Y_i = y \mid I_i = 1]}{P_{\mathcal{M}}[T^* = t^* \mid I_i = 1]}}{P_{\mathcal{M}}[Y_i \in \mathcal{S} \mid I_i = 1] \, P_{\mathcal{M}}[T^* = t^* \mid I_i = 0]} \tag{52}$$

$$= \frac{\sum_{y \in \mathcal{S}} \frac{P_{\mathcal{M}}[T^* = t^* \mid Y_i = y, I_i = 1]}{P_{\mathcal{M}}[T^* = t^* \mid I_i = 0]} \, P_{\mathcal{M}}[Y_i = y \mid I_i = 1]}{P_{\mathcal{M}}[Y_i \in \mathcal{S} \mid I_i = 1]} \tag{53}$$

We apply (44) to achieve the left bound in the first expression of (10).

$$\frac{P_{\mathcal{M}}[T^* = t^* \mid Y_i \in \mathcal{S}, I_i = 1]}{P_{\mathcal{M}}[T^* = t^* \mid I_i = 0]} \geq \frac{\sum_{y \in \mathcal{S}} e^{-\varepsilon} P_{\mathcal{M}}[Y_i = y \mid I_i = 1]}{P_{\mathcal{M}}[Y_i \in \mathcal{S} \mid I_i = 1]} = e^{-\varepsilon}. \tag{54}$$

We achieve the right bound in the first expression of (10) in a similar fashion.

$$\frac{P_{\mathcal{M}}[T^* = t^* \mid Y_i \in \mathcal{S}, I_i = 1]}{P_{\mathcal{M}}[T^* = t^* \mid I_i = 0]} \leq \frac{\sum_{y \in \mathcal{S}} e^{\varepsilon} P_{\mathcal{M}}[Y_i = y \mid I_i = 1]}{P_{\mathcal{M}}[Y_i \in \mathcal{S} \mid I_i = 1]} = e^{\varepsilon}. \tag{55}$$

We now turn to the second expression in (10). First note that

$$\frac{P_{\mathcal{M}}[T^* = t^* \mid Y_i \in \mathcal{S}, I_i = 1]}{P_{\mathcal{M}}[T^* = t^* \mid Y_i \notin \mathcal{S}, I_i = 1]}$$
$$= \frac{P_{\mathcal{M}}[T^* = t^* \mid Y_i \in \mathcal{S}, I_i = 1]}{P_{\mathcal{M}}[T^* = t^* \mid I_i = 0]} \cdot \frac{P_{\mathcal{M}}[T^* = t^* \mid I_i = 0]}{P_{\mathcal{M}}[T^* = t^* \mid Y_i \notin \mathcal{S}, I_i = 1]}. \tag{56}$$

By applying the bounds in the first expression in (10) to $\mathcal{S}$ and $\mathcal{S}^C$,

$$e^{-\varepsilon} \leq \frac{P_{\mathcal{M}}[T^* = t^* \mid Y_i \in \mathcal{S}, I_i = 1]}{P_{\mathcal{M}}[T^* = t^* \mid I_i = 0]} \leq e^{\varepsilon}, \tag{57}$$

$$e^{-\varepsilon} \leq \frac{P_{\mathcal{M}}[T^* = t^* \mid Y_i \notin \mathcal{S}, I_i = 1]}{P_{\mathcal{M}}[T^* = t^* \mid I_i = 0]} \leq e^{\varepsilon}. \tag{58}$$

Thus, to achieve the left inequality in the second expression in (10),

$$\frac{P_{\mathcal{M}}[T^* = t^* \mid Y_i \in \mathcal{S}, I_i = 1]}{P_{\mathcal{M}}[T^* = t^* \mid Y_i \notin \mathcal{S}, I_i = 1]} \geq e^{-\varepsilon} \cdot (e^{\varepsilon})^{-1} = e^{-2\varepsilon}. \tag{59}$$

To achieve the right inequality in the second expression in (10),

$$\frac{P_{\mathcal{M}}[T^* = t^* \mid Y_i \in \mathcal{S}, I_i = 1]}{P_{\mathcal{M}}[T^* = t^* \mid Y_i \notin \mathcal{S}, I_i = 1]} \leq e^{\varepsilon} \cdot \left(e^{-\varepsilon}\right)^{-1} = e^{2\varepsilon}. \tag{60}$$

$\square$

**Theorem 1.** *Under Assumption 1 and Assumption 2, if the release of $T^* = t^*$ satisfies DP, then*

$$r_i(p_i, q_i, t^*) \leq \frac{1}{q_i p_i + e^{-2\varepsilon}(1 - q_i)p_i + e^{-\varepsilon}(1 - p_i)}. \tag{11}$$

*Proof.* We begin by applying Bayes' Theorem to reverse the conditional in the relative risk.

$$r_i(p_i, q_i, t^*) = \frac{P_{\mathcal{M}}[Y_i \in \mathcal{S}, I_i = 1 \mid T^* = t^*]}{P_{\mathcal{M}}[Y_i \in \mathcal{S}, I_i = 1]} \tag{61}$$

$$= \frac{\frac{P_{\mathcal{M}}[T^* = t^* \mid Y_i \in \mathcal{S}, I_i = 1] \, P_{\mathcal{M}}[Y_i \in \mathcal{S}, I_i = 1]}{P_{\mathcal{M}}[T^* = t^*]}}{P_{\mathcal{M}}[Y_i \in \mathcal{S}, I_i = 1]} \tag{62}$$

$$= \frac{P_{\mathcal{M}}[T^* = t^* \mid Y_i \in \mathcal{S}, I_i = 1]}{P_{\mathcal{M}}[T^* = t^*]}. \tag{63}$$

We may decompose the denominator via the law of total probability.

$$
\begin{aligned}
P_{\mathcal{M}}[T^* = t^*] &= P_{\mathcal{M}}[T^* = t^* \mid Y_i \in \mathcal{S}, I_i = 1] \, P_{\mathcal{M}}[Y_i \in \mathcal{S} \mid I_i = 1] \, P_{\mathcal{M}}[I_i = 1] \\
&\quad + P_{\mathcal{M}}[T^* = t^* \mid Y_i \notin \mathcal{S}, I_i = 1] \, P_{\mathcal{M}}[Y_i \notin \mathcal{S} \mid I_i = 1] \, P_{\mathcal{M}}[I_i = 1] \\
&\quad + P_{\mathcal{M}}[T^* = t^* \mid I_i = 0] \, P_{\mathcal{M}}[I_i = 0] \\
&= P_{\mathcal{M}}[T^* = t^* \mid Y_i \in \mathcal{S}, I_i = 1] \, q_i p_i \\
&\quad + P_{\mathcal{M}}[T^* = t^* \mid Y_i \notin \mathcal{S}, I_i = 1] \, (1 - q_i)p_i \\
&\quad + P_{\mathcal{M}}[T^* = t^* \mid I_i = 0] \, (1 - p_i).
\end{aligned}
\tag{64}
$$
$$\tag{65}$$

Using this expansion in the expression for $r_i$ and dividing through by the numerator yields

$$r_i(p_i, q_i, t^*) = \frac{1}{q_i p_i + \frac{P_{\mathcal{M}}[T^* = t^* \mid Y_i \notin \mathcal{S}, I_i = 1]}{P_{\mathcal{M}}[T^* = t^* \mid Y_i \in \mathcal{S}, I_i = 1]}(1 - q_i)p_i + \frac{P_{\mathcal{M}}[T^* = t^* \mid I_i = 0]}{P_{\mathcal{M}}[T^* = t^* \mid Y_i \in \mathcal{S}, I_i = 1]}(1 - p_i)}. \tag{66}$$

Using Lemma 1, we then have

$$r_i(p_i, q_i, t^*) \leq \frac{1}{q_i p_i + e^{-2\varepsilon}(1 - q_i)p_i + e^{-\varepsilon}(1 - p_i)}. \tag{67}$$

$$\square$$

**Theorem 2.** *For any individual $i$, fix the prior probabilities, $p_i$ and $q_i$, and a desired bound on the relative disclosure risk, $r^*(p_i, q_i)$. Define $\varepsilon_i(p_i, q_i)$ to be the function.*

$$\varepsilon_i(p_i, q_i) = \begin{cases} \log\left(\dfrac{2p_i(1-q_i)}{\sqrt{(1-p_i)^2+4p_i(1-q_i)\left(\frac{1}{r^*(p_i,q_i)}-p_iq_i\right)}-(1-p_i)}\right), & \text{if } 0 < q_i < 1; \\[2em] \log\left(\dfrac{1-p_i}{\frac{1}{r^*(p_i,1)}-p_i}\right), & \text{if } q_i = 1. \end{cases} \tag{12}$$

*Under the conditions of Theorem 1, any statistic $T^* = t^*$ released under $\varepsilon$-DP with $\varepsilon \leq \varepsilon_i(p_i, q_i)$ will satisfy $r_i(p_i, q_i, t^*) \leq r^*(p_i, q_i)$.*

*Proof.* We begin with the simpler case where $q_i = 1$. By Theorem 1,

$$r_i(p_i, 1, t^*) \leq \frac{1}{p_i + e^{-\varepsilon}(1-p_i)}. \tag{68}$$

Since $\varepsilon \leq \log\left((1-p_i)/(1/r^*(p_i, 1) - p_i)\right)$, it follows that $e^{-\varepsilon} \geq (1/r^*(p_i, 1) - p_i)/(1-p_i)$. Thus, as desired,

$$r_i(p_i, 1, t^*) \leq \frac{1}{p_i + \frac{\frac{1}{r^*(p_i,1)}-p_i}{1-p_i}(1-p_i)} = \frac{1}{p_i + \left(\frac{1}{r^*(p_i,1)} - p_i\right)} = r^*(p_i, 1). \tag{69}$$

Now consider the case where $0 < q_i < 1$. By Theorem 1,

$$r_i(p_i, q_i, t^*) \leq \frac{1}{q_i p_i + e^{-2\varepsilon}(1-q_i)p_i + e^{-\varepsilon}(1-p_i)}. \tag{70}$$

Since

$$\varepsilon \leq \log\left(\frac{2p_i(1-q_i)}{\sqrt{(1-p_i)^2 + 4p_i(1-q_i)\left(\frac{1}{r^*(p_i,q_i)} - p_iq_i\right)} - (1-p_i)}\right), \tag{71}$$

it follows that

$$e^{-\varepsilon} \geq \frac{\sqrt{(1-p_i)^2 + 4p_i(1-q_i)\left(\frac{1}{r^*(p_i,q_i)} - p_iq_i\right)} - (1-p_i)}{2p_i(1-q_i)}. \tag{72}$$

Taking the square gives

$$e^{-2\varepsilon} \geq \frac{4p_i(1-q_i)\left(\frac{1}{r^*(p_i,q_i)} - p_iq_i\right) + 2(1-p_i)^2}{4p_i^2(1-q_i)^2}$$
$$- \frac{2(1-p_i)\sqrt{(1-p_i)^2 + 4p_i(1-q_i)\left(\frac{1}{r^*(p_i,q_i)} - p_iq_i\right)}}{4p_i^2(1-q_i)^2} \tag{73}$$

$$= \frac{\frac{1}{r^*(p_i,q_i)} - p_iq_i}{p_i(1-q_i)} + \frac{(1-p_i)^2 - (1-p_i)\sqrt{(1-p_i)^2 + 4p_i(1-q_i)\left(\frac{1}{r^*(p_i,q_i)} - p_iq_i\right)}}{2p_i^2(1-q_i)^2}. \tag{74}$$

Thus,

$$q_i p_i + e^{-2\varepsilon}(1-q_i)p_i + e^{-\varepsilon}(1-p_i)$$

$$\geq q_i p_i + \frac{\frac{1}{r^*(p_i,q_i)} - p_i q_i}{p_i(1-q_i)} \cdot (1-q_i)p_i$$

$$+ \frac{(1-p_i)^2 - (1-p_i)\sqrt{(1-p_i)^2 + 4p_i(1-q_i)\left(\frac{1}{r^*(p_i,q_i)} - p_i q_i\right)}}{2p_i^2(1-q_i)^2} \cdot (1-q_i)p_i$$

$$+ \frac{\sqrt{(1-p_i)^2 + 4p_i(1-q_i)\left(\frac{1}{r^*(p_i,q_i)} - p_i q_i\right)} - (1-p_i)}{2p_i(1-q_i)} \cdot (1-p_i) \qquad (75)$$

$$= q_i p_i + \left(\frac{1}{r^*(p_i,q_i)} - p_i q_i\right)$$

$$+ \left((1-p_i) - \sqrt{(1-p_i)^2 + 4p_i(1-q_i)\left(\frac{1}{r^*(p_i,q_i)} - p_i q_i\right)}\right)\frac{1-p_i}{2p_i(1-q_i)}$$

$$- \left((1-p_i) - \sqrt{(1-p_i)^2 + 4p_i(1-q_i)\left(\frac{1}{r^*(p_i,q_i)} - p_i q_i\right)}\right)\frac{1-p_i}{2p_i(1-q_i)} \qquad (76)$$

$$= \frac{1}{r^*(p_i,q_i)}. \qquad (77)$$

It is then immediate that

$$r_i(p_i, q_i, t^*) \leq \frac{1}{q_i p_i + e^{-2\varepsilon}(1-q_i)p_i + e^{-\varepsilon}(1-p_i)} \leq \frac{1}{\frac{1}{r^*(p_i,q_i)}} = r^*(p_i, q_i). \qquad (78)$$

$\square$

**Corollary 1.** *Under the conditions of Theorem 1, for all $p_i, q_i \in (0, 1]$ and all $t^*$,*

$$r_i(p_i, q_i, t^*) \leq e^{2\varepsilon}. \tag{79}$$

*Proof.* First note that since, $0 \leq e^{-\varepsilon} \leq 1$, it follows that $e^{-2\varepsilon} \leq e^{-\varepsilon}$. Then, applying the result of Theorem 1,

$$r_i(p_i, q_i, t^*) \leq \frac{1}{q_i p_i + e^{-2\varepsilon}(1 - q_i)p_i + e^{-\varepsilon}(1 - p_i)} \leq \frac{1}{q_i p_i + e^{-2\varepsilon}(1 - q_i)p_i + e^{-2\varepsilon}(1 - p_i)} \tag{80}$$

Combining terms and using the fact that $q_i p_i \geq 0$ gives

$$r_i(p_i, q_i, t^*) \leq \frac{1}{q_i p_i + e^{-2\varepsilon}(1 - q_i p_i)} \leq \frac{1}{e^{-2\varepsilon} + q_i p_i(1 - e^{-2\varepsilon})} \leq \frac{1}{e^{-2\varepsilon} + 0} = e^{2\varepsilon}. \tag{81}$$

$\square$

**Corollary 2.** *Under the conditions of Theorem 2, if $p_i = 1$ and $0 < q_i < 1$, then any statistic $T^* = t^*$ released under $\varepsilon$-DP with*

$$\varepsilon \leq \frac{1}{2} \log \left( \frac{1 - q_i}{\frac{1}{r^*(1, q_i)} - q_i} \right), \tag{82}$$

*will satisfy $r_i(1, q_i, t^*) \leq r^*(1, q_i)$.*

*Proof.* Plugging $p_i = 1$ into the expression from Theorem 2 yields

$$\varepsilon \leq \log \left( \frac{2(1 - q_i)}{\sqrt{0 + 4(1 - q_i)\left(\frac{1}{r^*(1, q_i)} - q_i\right)} - 0} \right) \tag{83}$$

$$= \log \left( \sqrt{\frac{1 - q_i}{\frac{1}{r^*(1, q_i)} - q_i}} \right) \tag{84}$$

$$= \frac{1}{2} \log \left( \frac{1 - q_i}{\frac{1}{r^*(1, q_i)} - q_i} \right). \tag{85}$$

$\square$

**Lemma 2.** *Fix $p_i, q_i \in (0, 1)$ and let $r^*(p_i, q_i) = \tilde{a}/(p_i q_i)$. Then the function*

$$\varepsilon(p_i, q_i) = \log \left( \frac{2 p_i (1 - q_i)}{\sqrt{(1-p_i)^2 + 4 p_i (1-q_i) \left( \frac{1}{r^*(p_i, q_i)} - p_i q_i \right)} - (1 - p_i)} \right) \tag{86}$$

*has partial derivatives such that*

1. $\frac{\partial \varepsilon(p_i, q_i)}{\partial p_i} < 0$ *for all $0 < p_i < 1$ and $0 < q_i < 1$*

2. $\frac{\partial \varepsilon(p_i, q_i)}{\partial q_i} < 0$ *for all $0 < p_i < 1$ and $0 < q_i < 1$.*

*Proof.* To begin, we re-express the function of interest in the form

$$\varepsilon(p_i, q_i) = \log \left( 2 p_i (1 - q_i) \right) - \log \left( \sqrt{(1-p_i)^2 + 4 p_i (1-q_i) \left( \frac{p_i q_i}{\tilde{a}} - p_i q_i \right)} - (1 - p_i) \right) \tag{87}$$

$$= \log \left( 2 p_i (1 - q_i) \right) - \log \left( \sqrt{(1-p_i)^2 + 4 p_i^2 q_i (1-q_i) \left( \frac{1}{\tilde{a}} - 1 \right)} - (1 - p_i) \right). \tag{88}$$

We first examine $\partial \varepsilon(p_i, q_i)/\partial p_i$. Taking the partial derivative with respect to $q_i$ gives

$$\frac{\partial \varepsilon(p_i, q_i)}{\partial p_i} = \frac{1}{p_i} - \frac{\frac{-2(1-p_i)+8p_i q_i(1-q_i)\left(\frac{1}{\tilde{a}}-1\right)}{2\sqrt{(1-p_i)^2+4p_i^2 q_i(1-q_i)\left(\frac{1}{\tilde{a}}-1\right)}} + 1}{\sqrt{(1-p_i)^2 + 4p_i^2 q_i(1-q_i)\left(\frac{1}{\tilde{a}}-1\right)} - (1-p_i)} \tag{89}$$

$$= \frac{1}{p_i} + \frac{\frac{(1-p_i)-4p_i q_i(1-q_i)\left(\frac{1}{\tilde{a}}-1\right)-\sqrt{(1-p_i)^2+4p_i^2 q_i(1-q_i)\left(\frac{1}{\tilde{a}}-1\right)}}{\sqrt{(1-p_i)^2+4p_i^2 q_i(1-q_i)\left(\frac{1}{\tilde{a}}-1\right)}}}{\sqrt{(1-p_i)^2 + 4p_i^2 q_i(1-q_i)\left(\frac{1}{\tilde{a}}-1\right)} - (1-p_i)} \tag{90}$$

$$= \frac{\frac{(1-p_i)^2+4p_i^2 q_i(1-q_i)\left(\frac{1}{\tilde{a}}-1\right)+p_i(1-p_i)-4p_i^2 q_i(1-q_i)\left(\frac{1}{\tilde{a}}-1\right)-[(1-p_i)-p_i]\sqrt{(1-p_i)^2+4p_i^2 q_i(1-q_i)\left(\frac{1}{\tilde{a}}-1\right)}}{\sqrt{(1-p_i)^2+4p_i^2 q_i(1-q_i)\left(\frac{1}{\tilde{a}}-1\right)}}}{p_i \left[ \sqrt{(1-p_i)^2 + 4p_i^2 q_i(1-q_i)\left(\frac{1}{\tilde{a}}-1\right)} - (1-p_i) \right]} \tag{91}$$

$$= \frac{\frac{[(1-p_i)+p_i](1-p_i)-\sqrt{(1-p_i)^2+4p_i^2 q_i(1-q_i)\left(\frac{1}{\tilde{a}}-1\right)}}{\sqrt{(1-p_i)^2+4p_i^2 q_i(1-q_i)\left(\frac{1}{\tilde{a}}-1\right)}}}{p_i \left[ \sqrt{(1-p_i)^2 + 4p_i^2 q_i(1-q_i)\left(\frac{1}{\tilde{a}}-1\right)} - (1-p_i) \right]} \tag{92}$$

$$= \frac{\frac{-\left[\sqrt{(1-p_i)^2+4p_i^2 q_i(1-q_i)\left(\frac{1}{\tilde{a}}-1\right)}-(1-p_i)\right]}{\sqrt{(1-p_i)^2+4p_i^2 q_i(1-q_i)\left(\frac{1}{\tilde{a}}-1\right)}}}{p_i \left[ \sqrt{(1-p_i)^2 + 4p_i^2 q_i(1-q_i)\left(\frac{1}{\tilde{a}}-1\right)} - (1-p_i) \right]} \tag{93}$$

$$= -\frac{1}{p_i \sqrt{(1-p_i)^2 + 4p_i^2 q_i(1-q_i)\left(\frac{1}{\tilde{a}}-1\right)}}. \tag{94}$$

Certainly, the denominator of (94) is positive. Thus, $\partial \varepsilon(p_i, q_i)/\partial p_i < 0$, as desired.

We now examine $\partial\varepsilon(p_i, q_i)/\partial q_i$. Taking the partial derivative with respect to $q_i$ gives

$$\frac{\partial\varepsilon(p_i,q_i)}{\partial q_i} = -\frac{1}{1-q_i} - \frac{\frac{4p_i^2\left(\frac{1}{\tilde a}-1\right)\frac{\partial}{\partial q_i}[q_i(1-q_i)]}{2\sqrt{(1-p_i)^2+4p_i^2q_i(1-q_i)\left(\frac{1}{\tilde a}-1\right)}}}{\sqrt{(1-p_i)^2+4p_i^2q_i(1-q_i)\left(\frac{1}{\tilde a}-1\right)}-(1-p_i)} \tag{95}$$

$$= -\frac{\sqrt{(1-p_i)^2+4p_i^2q_i(1-q_i)\left(\frac{1}{\tilde a}-1\right)}-(1-p_i)+\frac{2p_i^2\left(\frac{1}{\tilde a}-1\right)(1-q_i)\frac{\partial}{\partial q_i}[q_i(1-q_i)]}{\sqrt{(1-p_i)^2+4p_i^2q_i(1-q_i)\left(\frac{1}{\tilde a}-1\right)}}}{(1-q_i)\left[\sqrt{(1-p_i)^2+4^2q_i(1-q_i)\left(\frac{1}{\tilde a}-1\right)}-(1-p_i)\right]} \tag{96}$$

$$= -\frac{\frac{(1-p_i)^2+4p_i^2q_i(1-q_i)\left(\frac{1}{\tilde a}-1\right)-(1-p_i)\sqrt{(1-p_i)^2+4p_i^2q_i(1-q_i)\left(\frac{1}{\tilde a}-1\right)}+2p_i^2\left(\frac{1}{\tilde a}-1\right)(1-q_i)\frac{\partial}{\partial q_i}[q_i(1-q_i)]}{\sqrt{(1-p_i)^2+4p_i^2q_i(1-q_i)\left(\frac{1}{\tilde a}-1\right)}}}{(1-q_i)\left[\sqrt{(1-p_i)^2+4p_i^2q_i(1-q_i)\left(\frac{1}{\tilde a}-1\right)}-(1-p_i)\right]}. \tag{97}$$

Consider the three parts of (97). Since $\tilde a < 1$, we have that in the denominator of the numerator, $\sqrt{(1-p_i)^2+4p_i^2q_i(1-q_i)\left(\frac{1}{\tilde a}-1\right)} > 0$. The denominator is also positive, since $(1-q_i) > 0$ and

$$\left[\sqrt{(1-p_i)^2+4p_i^2q_i(1-q_i)\left(\frac{1}{\tilde a}-1\right)}-(1-p_i)\right] > \left[\sqrt{(1-p_i)^2}-(1-p_i)\right] = 0. \tag{98}$$

This leaves the numerator of the numerator of (97). Let us denote this expression as $g(q_i)$. If $g(q_i) > 0$, then it follows that $\partial\varepsilon(p_i, q_i)/\partial q_i < 0$. To show this, we begin by simplifying the expression for $g(q_i)$:

$$g(q_i) = (1-p_i)^2 + 4p_i^2q_i(1-q_i)\left(\frac{1}{\tilde a}-1\right) - (1-p_i)\sqrt{(1-p_i)^2+4p_i^2q_i(1-q_i)\left(\frac{1}{\tilde a}-1\right)}$$
$$+ 2p_i^2\left(\frac{1}{\tilde a}-1\right)(1-q_i)\frac{\partial}{\partial q_i}[q_i(1-q_i)] \tag{99}$$

$$= (1-p_i)^2 + 2p_i^2(1-q_i)(2q_i)\left(\frac{1}{\tilde a}-1\right) - (1-p_i)\sqrt{(1-p_i)^2+4p_i^2(1-q_i)q_i\left(\frac{1}{\tilde a}-1\right)}$$
$$+ 2p_i^2(1-q_i)\left(\frac{1}{\tilde a}-1\right)(1-2q_i) \tag{100}$$

$$= (1-p_i)^2 - (1-p_i)\sqrt{(1-p_i)^2+4p_i^2(1-q_i)q_i\left(\frac{1}{\tilde a}-1\right)} + 2p_i^2(1-q_i)\left(\frac{1}{\tilde a}-1\right). \tag{101}$$

Taking the derivative of $g$, we find that

$$\frac{\partial g(q_i)}{\partial q_i} = -(1-p_i)\frac{4p_i^2(1-2q_i)\left(\frac{1}{\tilde a}-1\right)}{2\sqrt{(1-p_i)^2+4p_i^2(1-q_i)q_i\left(\frac{1}{\tilde a}-1\right)}} - 2p_i^2\left(\frac{1}{\tilde a}-1\right) \tag{102}$$

$$= -2p_i^2\left(\frac{1}{\tilde a}-1\right)\left[(1-p_i)\frac{1-2q_i}{\sqrt{(1-p_i)^2+4p_i^2(1-q_i)q_i\left(\frac{1}{\tilde a}-1\right)}}+1\right]. \tag{103}$$

For $0 < q_i \leq 1/2$, $1 - 2q_i \geq 0$ and so $\partial g(q_i)/\partial q_i < 0$. For $1/2 < q_i < 1$, since $2q_i - 1 < 1$ and $\sqrt{(1-p_i)^2 + 4p_i^2(1-q_i)q_i(1/\tilde{a} - 1)} > 1 - p_i$,

$$\frac{\partial g(q_i)}{\partial q_i} = -2p_i^2 \left(\frac{1}{\tilde{a}} - 1\right) \left[1 - (1-p_i)\frac{2q_i - 1}{\sqrt{(1-p_i)^2 + 4p_i^2(1-q_i)q_i\left(\frac{1}{\tilde{a}} - 1\right)}}\right] \tag{104}$$

$$< -2p_i^2 \left(\frac{1}{\tilde{a}} - 1\right) \left[1 - (1-p_i)\frac{1}{1 - p_i}\right] \tag{105}$$

$$< 0. \tag{106}$$

Thus, $g(q_i)$ strictly decreasing as a function of $q_i$ on $0 < q_i < 1$. Since

$$g(1) = (1-p_i)^2 - (1-p_i)\sqrt{(1-p_i)^2 + 4p_i^2(1-1)1\left(\frac{1}{\tilde{a}} - 1\right)} + 2p_i^2(1-1)\left(\frac{1}{\tilde{a}} - 1\right) = 0, \tag{107}$$

it follows that $g(q_i) > 0$ for $0 < q_i < 1$ and so $\partial \varepsilon(p_i, q_i)/\partial q_i < 0$ in this range. $\qquad \square$

**Lemma 3.** *Fix $p_i, q_i \in (0,1)$ and let $r^*(p_i, q_i) = \tilde{r} < 1/(p_i q_i)$. Then the function*

$$\varepsilon(p_i, q_i) = \log\left( \frac{2p_i(1-q_i)}{\sqrt{(1-p_i)^2 + 4p_i(1-q_i)\left(\frac{1}{r^*(p_i,q_i)} - p_i q_i\right)} - (1-p_i)} \right) \tag{108}$$

*has partial derivatives such that*

1. $\frac{\partial \varepsilon(p_i, q_i)}{\partial p_i} < 0$ *if* $0 < q_i < \frac{1}{\tilde{r}+1}$ *and* $0 < p_i < 1$

2. $\frac{\partial \varepsilon(p_i, q_i)}{\partial p_i} = 0$ *if* $q_i = \frac{1}{\tilde{r}+1}$ *and* $0 < p_i < 1$

3. $\frac{\partial \varepsilon(p_i, q_i)}{\partial p_i} > 0$ *if* $\frac{1}{\tilde{r}+1} < q_i < 1$ *and* $0 < p_i < \frac{1}{q_i \tilde{r}}$

4. $\frac{\partial \varepsilon(p_i, q_i)}{\partial q_i} > 0$ *if* $0 < p_i < 1$ *and* $0 < q_i < \frac{1}{p_i \tilde{r}}$.

*Proof.* To begin, we re-express the function of interest in the form

$$\varepsilon(p_i, q_i) = \log\left(2p_i(1-q_i)\right) - \log\left( \sqrt{(1-p_i)^2 + 4p_i(1-q_i)\left(\frac{1}{\tilde{r}} - p_i q_i\right)} - (1-p_i) \right). \tag{109}$$

We now examine $\partial \varepsilon(p_i, q_i)/\partial p_i$. Taking the partial derivative of with respect to $p_i$ and simplifying gives

$$\frac{\partial \varepsilon(p_i, q_i)}{\partial p_i} = \frac{1}{p_i} - \frac{\frac{\frac{\partial}{\partial p_i}\left[(1-p_i)^2 + 4p_i(1-q_i)\left(\frac{1}{\tilde{r}} - p_i q_i\right)\right]}{2\sqrt{(1-p_i)^2 + 4p_i(1-q_i)\left(\frac{1}{\tilde{r}} - p_i q_i\right)}} + 1}{\sqrt{(1-p_i)^2 + 4p_i(1-q_i)\left(\frac{1}{\tilde{r}} - p_i q_i\right)} - (1-p_i)} \tag{110}$$

$$= \frac{1}{p_i} - \frac{\frac{-2(1-p_i) + 4(1-q_i)\left(\frac{1}{\tilde{r}} - p_i q_i\right) - 4p_i(1-q_i)q_i + 2\sqrt{(1-p_i)^2 + 4p_i(1-q_i)\left(\frac{1}{\tilde{r}} - p_i q_i\right)}}{2\sqrt{(1-p_i)^2 + 4p_i(1-q_i)\left(\frac{1}{\tilde{r}} - p_i q_i\right)}}}{\sqrt{(1-p_i)^2 + 4p_i(1-q_i)\left(\frac{1}{\tilde{r}} - p_i q_i\right)} - (1-p_i)} \tag{111}$$

$$= \frac{\frac{(1-p_i)^2 + 4p_i(1-q_i)\left(\frac{1}{\tilde{r}} - p_i q_i\right) - (1-p_i)\sqrt{(1-p_i)^2 + 4p_i(1-q_i)\left(\frac{1}{\tilde{r}} - p_i q_i\right)}}{\sqrt{(1-p_i)^2 + 4p_i(1-q_i)\left(\frac{1}{\tilde{r}} - p_i q_i\right)}}}{p_i\left[\sqrt{(1-p_i)^2 + 4p_i(1-q_i)\left(\frac{1}{\tilde{r}} - p_i q_i\right)} - (1-p_i)\right]}$$

$$+ \frac{\frac{p_i(1-p_i) - 2p_i(1-q_i)\left(\frac{1}{\tilde{r}} - p_i q_i\right) + 2p_i^2(1-q_i)q_i - p_i\sqrt{(1-p_i)^2 + 4p_i(1-q_i)\left(\frac{1}{\tilde{r}} - p_i q_i\right)}}{\sqrt{(1-p_i)^2 + 4p_i(1-q_i)\left(\frac{1}{\tilde{r}} - p_i q_i\right)}}}{p_i\left[\sqrt{(1-p_i)^2 + 4p_i(1-q_i)\left(\frac{1}{\tilde{r}} - p_i q_i\right)} - (1-p_i)\right]} \tag{112}$$

$$= \frac{\frac{(1-p_i)^2 + 4p_i(1-q_i)\left(\frac{1}{\tilde{r}} - p_i q_i\right) + p_i(1-p_i) - 2p_i(1-q_i)\left(\frac{1}{\tilde{r}} - p_i q_i\right) + 2p_i^2(1-q_i)q_i - \sqrt{(1-p_i)^2 + 4p_i(1-q_i)\left(\frac{1}{\tilde{r}} - p_i q_i\right)}}{\sqrt{(1-p_i)^2 + 4p_i(1-q_i)\left(\frac{1}{\tilde{r}} - p_i q_i\right)}}}{p_i\left[\sqrt{(1-p_i)^2 + 4p_i(1-q_i)\left(\frac{1}{\tilde{r}} - p_i q_i\right)} - (1-p_i)\right]} \tag{113}$$

$$= \frac{2p_i(1-q_i)\frac{1}{\tilde{r}} + (1-p_i) - \sqrt{(1-p_i)^2 + 4p_i(1-q_i)\left(\frac{1}{\tilde{r}} - p_i q_i\right)}}{p_i\left[\sqrt{(1-p_i)^2 + 4p_i(1-q_i)\left(\frac{1}{\tilde{r}} - p_i q_i\right)} - (1-p_i)\right]\sqrt{(1-p_i)^2 + 4p_i(1-q_i)\left(\frac{1}{\tilde{r}} - p_i q_i\right)}}. \tag{114}$$

We can rearrange the expression in (114) to the following form.

$$\frac{\partial \varepsilon(p_i, q_i)}{\partial p_i} = \frac{\frac{2p_i(1-q_i)\frac{1}{\tilde{r}}}{\sqrt{(1-p_i)^2 + 4p_i(1-q_i)\left(\frac{1}{\tilde{r}} - p_i q_i\right)} - (1-p_i)} - 1}{p_i\sqrt{(1-p_i)^2 + 4p_i(1-q_i)\left(\frac{1}{\tilde{r}} - p_i q_i\right)}}. \tag{115}$$

The denominator of (115) is certainly always positive, so the sign of $\partial\varepsilon(p_i, q_i)/\partial p_i$ is determined by the sign of the numerator. To determine the sign, we will use the following equality.

$$2p_i(1 - q_i)\frac{1}{\tilde{r}}$$
$$= \sqrt{(1 - p_i)^2 + 4p_i(1 - q_i)\left(\frac{1}{\tilde{r}} - p_i q_i\right) + 4p_i^2(1 - q_i)\frac{\tilde{r}^2 - 1}{\tilde{r}^2}\left(q_i - \frac{1}{\tilde{r} + 1}\right)} - (1 - p_i). \tag{116}$$

To demonstrate that (116) holds, note the following.

$$\left(2p_i(1 - q_i)\frac{1}{\tilde{r}} + (1 - p_i)\right)^2 = 4p_i^2(1 - q_i)^2\frac{1}{\tilde{r}^2} + 4p_i(1 - p_i)(1 - q_i)\frac{1}{\tilde{r}} + (1 - p_i)^2 \tag{117}$$

$$= 4p_i^2(1 - q_i)^2\frac{1}{\tilde{r}^2} + 4p_i(1 - p_i)(1 - q_i)\frac{1}{\tilde{r}} - 4p_i^2(1 - q_i)\frac{\tilde{r}^2 - 1}{\tilde{r}^2}\left(q_i - \frac{1}{\tilde{r} + 1}\right)$$
$$+ 4p_i^2(1 - q_i)\frac{\tilde{r}^2 - 1}{\tilde{r}^2}\left(q_i - \frac{1}{\tilde{r} + 1}\right) + (1 - p_i)^2 \tag{118}$$

$$= 4p_i(1 - q_i)\left[p_i(1 - q_i)\frac{1}{\tilde{r}^2} + (1 - p_i)\frac{1}{\tilde{r}} - p_i\frac{\tilde{r}^2 - 1}{\tilde{r}^2}\left(q_i - \frac{1}{\tilde{r} + 1}\right)\right]$$
$$+ 4p_i^2(1 - q_i)\frac{\tilde{r}^2 - 1}{\tilde{r}^2}\left(q_i - \frac{1}{\tilde{r} + 1}\right) + (1 - p_i)^2 \tag{119}$$

$$= 4p_i(1 - q_i)\left[\frac{1}{\tilde{r}} + p_i\left(\frac{1}{\tilde{r}^2} - \frac{1}{\tilde{r}} + \frac{\tilde{r} - 1}{\tilde{r}^2}\right) - p_i q_i\left(\frac{1}{\tilde{r}^2} + \frac{\tilde{r}^2 - 1}{\tilde{r}^2}\right)\right]$$
$$+ 4p_i^2(1 - q_i)\frac{\tilde{r}^2 - 1}{\tilde{r}^2}\left(q_i - \frac{1}{\tilde{r} + 1}\right) + (1 - p_i)^2 \tag{120}$$

$$= 4p_i(1 - q_i)\left(\frac{1}{\tilde{r}} - p_i q_i\right) + 4p_i^2(1 - q_i)\frac{\tilde{r}^2 - 1}{\tilde{r}^2}\left(q_i - \frac{1}{\tilde{r} + 1}\right) + (1 - p_i)^2. \tag{121}$$

Rearranging (121) gives (116).

Now note that, from (116), if $q_i = 1/(\tilde{r} + 1)$, then

$$2p_i(1 - q_i)\frac{1}{\tilde{r}} = \sqrt{(1 - p_i)^2 + 4p_i(1 - q_i)\left(\frac{1}{\tilde{r}} - p_i q_i\right)} - (1 - p_i). \tag{122}$$

It follows that the ratio in the numerator of (115) equals 1 and thus $\partial\varepsilon(p_i, q_i)/\partial p_i = 0$. If $0 < q_i < \frac{1}{\tilde{r}+1}$, then the term $4p_i^2(1 - q_i)(\tilde{r}^2 - 1)/\tilde{r}^2(q_i - 1/(\tilde{r} + 1)) < 0$ and so

$$2p_i(1 - q_i)\frac{1}{\tilde{r}} < \sqrt{(1 - p_i)^2 + 4p_i(1 - q_i)\left(\frac{1}{\tilde{r}} - p_i q_i\right)} - (1 - p_i). \tag{123}$$

It follows that the ratio in the numerator of (115) is less than 1 and thus $\partial\varepsilon(p_i, q_i)/\partial p_i < 0$. If $\frac{1}{\tilde{r}+1} < q_i < 1$, then the term $4p_i^2(1 - q_i)(\tilde{r}^2 - 1)/\tilde{r}^2(q_i - 1/(\tilde{r} + 1)) > 0$ and so

$$2p_i(1 - q_i)\frac{1}{\tilde{r}} > \sqrt{(1 - p_i)^2 + 4p_i(1 - q_i)\left(\frac{1}{\tilde{r}} - p_i q_i\right)} - (1 - p_i). \tag{124}$$

It follows that the ratio in the numerator of (115) is greater than 1 and thus $\partial\varepsilon(p_i, q_i)/\partial p_i > 0$.

We now examine $\partial\varepsilon(p_i,q_i)/\partial q_i$. Taking the partial derivative of with respect to $q_i$ gives

$$\frac{\partial\varepsilon(p_i,q_i)}{\partial q_i} = -\frac{1}{1-q_i} - \frac{\frac{4p_i\frac{\partial}{\partial q_i}\left[(1-q_i)\left(\frac{1}{\tilde{r}}-p_iq_i\right)\right]}{2\sqrt{(1-p_i)^2+4p_i(1-q_i)\left(\frac{1}{\tilde{r}}-p_iq_i\right)}}}{\sqrt{(1-p_i)^2+4p_i(1-q_i)\left(\frac{1}{\tilde{r}}-p_iq_i\right)}-(1-p_i)} \tag{125}$$

$$= -\frac{\sqrt{(1-p_i)^2+4p_i(1-q_i)\left(\frac{1}{\tilde{r}}-p_iq_i\right)}-(1-p_i)+\frac{2p_i(1-q_i)\frac{\partial}{\partial q_i}\left[(1-q_i)\left(\frac{1}{\tilde{r}}-p_iq_i\right)\right]}{\sqrt{(1-p_i)^2+4p_i(1-q_i)\left(\frac{1}{\tilde{r}}-p_iq_i\right)}}}{(1-q_i)\left[\sqrt{(1-p_i)^2+4p_i(1-q_i)\left(\frac{1}{\tilde{r}}-p_iq_i\right)}-(1-p_i)\right]} \tag{126}$$

$$= -\frac{\frac{(1-p_i)^2+4p_i(1-q_i)\left(\frac{1}{\tilde{r}}-p_iq_i\right)-(1-p_i)\sqrt{(1-p_i)^2+4p_i(1-q_i)\left(\frac{1}{\tilde{r}}-p_iq_i\right)}+2p_i(1-q_i)\frac{\partial}{\partial q_i}\left[(1-q_i)\left(\frac{1}{\tilde{r}}-p_iq_i\right)\right]}{\sqrt{(1-p_i)^2+4p_i(1-q_i)\left(\frac{1}{\tilde{r}}-p_iq_i\right)}}}{(1-q_i)\left[\sqrt{(1-p_i)^2+4p_i(1-q_i)\left(\frac{1}{\tilde{r}}-p_iq_i\right)}-(1-p_i)\right]}. \tag{127}$$

Consider the three parts of (127). Since $\tilde{r} < 1/(p_iq_i)$, it follows that $(1/\tilde{r}-p_iq_i) > 0$. Thus, in the denominator of the numerator, $\sqrt{(1-p_i)^2+4p_i(1-q_i)(1/\tilde{r}-p_iq_i)} > 0$. The denominator is also positive, since $(1-q_i) > 0$ and

$$\left[\sqrt{(1-p_i)^2+4p_i(1-q_i)\left(\frac{1}{\tilde{r}}-p_iq_i\right)}-(1-p_i)\right] > \left[\sqrt{(1-p_i)^2}-(1-p_i)\right] = 0. \tag{128}$$

This leaves the numerator of the numerator of (127). Let us denote this expression as $g(q_i)$. If $g(q_i) < 0$, then it follows that $\partial\varepsilon(p_i,q_i)/\partial q_i > 0$. To show this, we begin by simplifying the expression for $g(q_i)$:

$$g(q_i) = (1-p_i)^2 + 4p_i(1-q_i)\left(\frac{1}{\tilde{r}}-p_iq_i\right) - (1-p_i)\sqrt{(1-p_i)^2+4p_i(1-q_i)\left(\frac{1}{\tilde{r}}-p_iq_i\right)}$$

$$+ 2p_i(1-q_i)\frac{\partial}{\partial q_i}\left[(1-q_i)\left(\frac{1}{\tilde{r}}-p_iq_i\right)\right] \tag{129}$$

$$= (1-p_i)^2 + 2p_i(1-q_i)\left(\frac{2}{\tilde{r}}-2p_iq_i\right) - (1-p_i)\sqrt{(1-p_i)^2+4p_i(1-q_i)\left(\frac{1}{\tilde{r}}-p_iq_i\right)}$$

$$+ 2p_i(1-q_i)\left(2p_iq_i-p_i-\frac{1}{\tilde{r}}\right) \tag{130}$$

$$= (1-p_i)^2 - (1-p_i)\sqrt{(1-p_i)^2+4p_i(1-q_i)\left(\frac{1}{\tilde{r}}-p_iq_i\right)} + 2p_i(1-q_i)\left(\frac{1}{\tilde{r}}-p_i\right). \tag{131}$$

Taking the derivative of $g$, we find that

$$\frac{\partial g(q_i)}{\partial q_i} = -(1-p_i)\frac{4p_i\left(2p_iq_i-p_i-\frac{1}{\tilde{r}}\right)}{2\sqrt{(1-p_i)^2+4p_i(1-q_i)\left(\frac{1}{\tilde{r}}-p_iq_i\right)}} - 2p_i\left(\frac{1}{\tilde{r}}-p_i\right) \tag{132}$$

$$= 2p_i\left[(1-p_i)\frac{2p_i(1-q_i)+\left(\frac{1}{\tilde{r}}-p_i\right)}{\sqrt{(1-p_i)^2+4p_i(1-q_i)\left(\frac{1}{\tilde{r}}-p_iq_i\right)}} - \left(\frac{1}{\tilde{r}}-p_i\right)\right]. \tag{133}$$

Note that since $\tilde{r} < \frac{1}{p_i q_i}$, it follows that $\left(\frac{1}{\tilde{r}} - p_i q_i\right) > 0$ and $p_i(1 - q_i) > p_i - \frac{1}{r}$. Thus,

$$\frac{\partial g(q_i)}{\partial q_i} = 2p_i \left[ (1 - p_i) \frac{2p_i(1 - q_i) - \left(p_i - \frac{1}{\tilde{r}}\right)}{\sqrt{(1 - p_i)^2 + 4p_i(1 - q_i)\left(\frac{1}{\tilde{r}} - p_i q_i\right)}} + \left(p_i - \frac{1}{\tilde{r}}\right) \right] \tag{134}$$

$$> 2p_i \left[ (1 - p_i) \frac{\left(p_i - \frac{1}{\tilde{r}}\right)}{\sqrt{(1 - p_i)^2}} + \left(p_i - \frac{1}{\tilde{r}}\right) \right] \tag{135}$$

$$= 4p_i \left(p_i - \frac{1}{\tilde{r}}\right) \tag{136}$$

$$> 0. \tag{137}$$

Thus, $g(q_i)$ is strictly increasing for $\tilde{r} < 1/(p_i q_i)$. Note that in terms of $q_i$, this is equivalent to the range $0 < q_i < 1/(p_i \tilde{r})$. Since

$$g\left(\frac{1}{p_i \tilde{r}}\right)$$

$$= (1 - p_i)^2 - (1 - p_i)\sqrt{(1 - p_i)^2 + 4p_i(1 - \frac{1}{p_i \tilde{r}})\left(\frac{1}{\tilde{r}} - p_i \frac{1}{p_i \tilde{r}}\right)} + 2p_i\left(1 - \frac{1}{p_i \tilde{r}}\right)\left(\frac{1}{\tilde{r}} - p_i\right) \tag{138}$$

$$= 2p_i \left(1 - \frac{1}{p_i \tilde{r}}\right)\left(\frac{1}{\tilde{r}} - p_i\right) \tag{139}$$

$$< 0, \tag{140}$$

it follows that $g(q_i) < 0$ for $0 < q_i < 1/(p_i \tilde{r})$. Thus, as desired, $\partial \varepsilon(p_i, q_i)/\partial q_i > 0$ in this range. $\qquad \square$

**Theorem 3.** *Under the conditions of Theorem 2, if $r^*(p_i, q_i) = \tilde{r} > 1$, the solution to the minimization problem in (13) is*

$$\varepsilon = \frac{1}{2} \log\left(\tilde{r}\right). \tag{141}$$

*Proof.* We define $\varepsilon(p_i, q_i)$ as follows.

$$\varepsilon(p_i, q_i) = \begin{cases} \log\left(\dfrac{2p_i(1-q_i)}{\sqrt{(1-p_i)^2 + 4p_i(1-q_i)\left(\frac{1}{\tilde{r}} - p_i q_i\right)} - (1-p_i)}\right), & \text{if } 0 < q_i < 1; \\ \log\left(\dfrac{1-p_i}{\frac{1}{\tilde{r}} - p_i}\right), & \text{if } q_i = 1. \end{cases} \tag{142}$$

The solution to the minimization problem in (13) corresponds to the minimum of $\varepsilon(p_i, q_i)$ over $(0,1] \times (0,1]$. Lemma 3 implies that $\partial\varepsilon(p_i, q_i)/\partial q_i \neq 0$ for any $0 < q_i < 1$. This means that this function must take its minimum value around its boundary, i.e., when $p_i = 1$, $q_i = 1$, $p_i \to 0$, or $q_i \to 0$. We examine each of these in turn.

We begin with the boundary where $p_i = 1$. By Corollary 2, for $0 < q_i < 1$,

$$\varepsilon(1, q_i) = \frac{1}{2} \log\left(\frac{1-q_i}{\frac{1}{\tilde{r}} - q_i}\right). \tag{143}$$

We take the partial derivative of this function with respect to $q_i$.

$$\frac{\partial\varepsilon(1, q_i)}{\partial q_i} = \frac{1}{2}\left[-\frac{1}{1-q_i} + \frac{1}{\frac{1}{\tilde{r}} - q_i}\right]. \tag{144}$$

Since $1/\tilde{r} < 1$, it follows that $\partial\varepsilon(1, q_i)/\partial q_i > 0$ for all $0 < q_i < 1$. Thus, the minimum occurs as $q_i \to 0$ and is

$$\lim_{q_i \to 0} \varepsilon(1, q_i) = \frac{1}{2}\log(\tilde{r}). \tag{145}$$

We next examine with the boundary where $q_i = 1$. When $0 < p_i < 1$,

$$\varepsilon(p_i, 1) = \log\left(\frac{1-p_i}{\frac{1}{\tilde{r}} - p_i}\right). \tag{146}$$

We take the partial derivative of this function with respect to $p_i$.

$$\frac{\partial\varepsilon(p_i, 1)}{\partial p_i} = -\frac{1}{1-p_i} + \frac{1}{\frac{1}{\tilde{r}} - p_i}. \tag{147}$$

Since $1/\tilde{r} < 1$, it follows that $\partial\varepsilon(p_i, 1)/\partial p_i > 0$ for all $0 < p_i < 1$. Thus, the minimum occurs as $p_i \to 0$ and is

$$\lim_{p_i \to 0} \varepsilon(p_i, 1) = \log(\tilde{r}). \tag{148}$$

We next examine the boundary where $p_i \to 0$. Since the logarithm is a continuous function, for all $0 < q_i < 1$,

$$\lim_{p_i \to 0} \varepsilon(p_i, q_i) = \lim_{p_i \to 0} \log\left(\frac{2p_i(1-q_i)}{\sqrt{(1-p_i)^2 + 4p_i(1-q_i)\left(\frac{1}{\tilde{r}} - p_i q_i\right)} - (1-p_i)}\right) \tag{149}$$

$$= \log\left(\lim_{p_i \to 0} \frac{2p_i(1-q_i)}{\sqrt{(1-p_i)^2 + 4p_i(1-q_i)\left(\frac{1}{\tilde{r}} - p_i q_i\right)} - (1-p_i)}\right). \tag{150}$$

Using L'Hôpital's Rule,

$$\lim_{p_i \to 0} \varepsilon(p_i, q_i) = \log\left(\lim_{p_i \to 0} \frac{2(1-q_i)}{\frac{-2(1-p_i) + 4(1-q_i)\left(\frac{1}{\tilde{r}} - p_i q_i\right) - 4p_i q_i(1-q_i)}{2\sqrt{(1-p_i)^2 + 4p_i(1-q_i)\left(\frac{1}{\tilde{r}} - p_i q_i\right)}} + 1}\right) \tag{151}$$

$$= \log\left(\frac{2(1-q_i)}{\frac{-1 + 2(1-q_i)\left(\frac{1}{\tilde{r}}\right)}{\sqrt{1}} + 1}\right) = \log(\tilde{r}). \tag{152}$$

Finally, we examine the boundary where $q_i \to 0$. For all $0 < p_i \leq 1$

$$\lim_{q_i \to 0} \varepsilon(p_i, q_i) = \lim_{q_i \to 0} \log \left( \frac{2p_i(1 - q_i)}{\sqrt{(1 - p_i)^2 + 4p_i(1 - q_i)\left(\frac{1}{\tilde{r}} - p_i q_i\right)} - (1 - p_i)} \right) \tag{153}$$

$$= \log \left( \frac{2p_i}{\sqrt{(1 - p_i)^2 + 4p_i\left(\frac{1}{\tilde{r}}\right)} - (1 - p_i)} \right). \tag{154}$$

We take the partial derivative with respect to $p_i$. From (115) in the proof of Lemma 3, we have that

$$\frac{\partial}{\partial p_i} \left( \lim_{q_i \to 0} \varepsilon(p_i, q_i) \right) = \frac{\partial}{\partial p_i} \left( \log \left( \frac{2p_i}{\sqrt{(1 - p_i)^2 + 4p_i\left(\frac{1}{\tilde{r}} - p_i\right)} - (1 - p_i)} \right) \right) \tag{155}$$

$$= \frac{\frac{2p_i \frac{1}{\tilde{r}}}{\sqrt{(1 - p_i)^2 + 4p_i\left(\frac{1}{\tilde{r}}\right)} - (1 - p_i)} - 1}{p_i \sqrt{(1 - p_i)^2 + 4p_i\left(\frac{1}{\tilde{r}}\right)}}. \tag{156}$$

By (116) with $q_i = 0$, the ratio in the numerator of (156) is less than 1 and so $\partial/\partial p_i \left( \lim_{q_i \to 0} \varepsilon(p_i, q_i) \right)$ is negative for all $0 < p_i \leq 1$. Thus, the minimum occurs at $p_i = 1$ and is as in (145). The global minimum of $\varepsilon(p_i, q_i)$ is thus $1/2 \log(\tilde{r})$. $\qquad \square$

**Theorem 4.** *Under the conditions of Theorem 2, let $\tilde{a} < 1$, $\tilde{p} \leq 1$, and $\tilde{r} > 1$, and $0 < q_i \leq 1$. If the function $r^*$ is such that $r^*(\tilde{p}, q_i) = \max\{\tilde{a}/(\tilde{p}q_i), \tilde{r}\}$ and $r^*(p_i, q_i) = \infty$ if $p_i \neq \tilde{p}$, then the solution to the minimization problem in (13) is*

$$
\varepsilon = \begin{cases} \log\left(\frac{\tilde{a}(1-\tilde{p})}{\tilde{p}(1-\tilde{a})}\right), & \text{if } 0 < \tilde{p} \leq \frac{\tilde{a}}{\tilde{r}}; \\ \log\left(\frac{2(\tilde{p}\tilde{r}-\tilde{a})}{\sqrt{\tilde{r}^2(1-\tilde{p})^2+4(\tilde{p}\tilde{r}-\tilde{a})(1-\tilde{a})}-\tilde{r}(1-\tilde{p})}\right), & \text{if } \frac{\tilde{a}}{\tilde{r}} < \tilde{p} \leq 1. \end{cases} \tag{157}
$$

*Proof.* We define $\varepsilon(q_i)$ as follows.

$$
\varepsilon(q_i) = \begin{cases} \log\left(\frac{2\tilde{p}(1-q_i)}{\sqrt{(1-\tilde{p})^2+4\tilde{p}(1-q_i)\left(\frac{1}{r^*(\tilde{p},q_i)}-\tilde{p}q_i\right)}-(1-\tilde{p})}\right), & \text{if } 0 < q_i < 1; \\ \log\left(\frac{1-\tilde{p}}{\frac{1}{r^*(\tilde{p},q_i)}-\tilde{p}}\right), & \text{if } q_i = 1. \end{cases} \tag{158}
$$

The solution to the minimization problem in (13) corresponds to the minimum of $\varepsilon(q_i)$ over $(0, 1]$.

We begin with the case where $0 < \tilde{p} \leq \tilde{a}/\tilde{r}$. In this setting, we have that $\tilde{r} \leq \tilde{a}/\tilde{p} \leq \tilde{a}/(\tilde{p}q_i)$, so $r^*(\tilde{p}, q_i) = \tilde{a}/(\tilde{p}q_i)$ for all $q_i$. By Lemma 2, $\varepsilon(q_i)$ is a decreasing function of $q_i$, so the minimum occurs at $q_i = 1$ and is

$$
\varepsilon(1) = \log\left(\frac{1-\tilde{p}}{\frac{1}{r^*(\tilde{p},1)}-\tilde{p}}\right) = \log\left(\frac{1-\tilde{p}}{\frac{\tilde{p}}{\tilde{a}}-\tilde{p}}\right) = \log\left(\frac{\tilde{a}(1-\tilde{p})}{\tilde{p}(1-\tilde{a})}\right). \tag{159}
$$

We now consider the case where $\tilde{a}/\tilde{r} < \tilde{p} \leq 1$. Note that $\tilde{r} \geq \tilde{a}/(\tilde{p}q_i)$ whenever $q_i \geq \tilde{a}/(\tilde{p}\tilde{r})$. Since in this setting $\tilde{a}/(\tilde{p}\tilde{r}) < 1$, the risk bound can be written in piece-wise form

$$
r^*(\tilde{p}, q_i) = \begin{cases} \frac{\tilde{a}}{\tilde{p}q_i}, & \text{if } 0 < q_i < \frac{\tilde{a}}{\tilde{p}\tilde{r}}; \\ \tilde{r}, & \text{if } \frac{\tilde{a}}{\tilde{p}\tilde{r}} \leq q_i < 1. \end{cases} \tag{160}
$$

Note that if $q_i \geq 1/(\tilde{r}\tilde{p})$, then $\tilde{r} \geq 1/(\tilde{p}q_i)$. Thus, $r^*(\tilde{p}, q_i) \geq \tilde{r} \geq 1/(\tilde{p}q_i)$ and so

$$
\sqrt{(1-\tilde{p})^2 + 4\tilde{p}(1-q_i)\left(\frac{1}{r^*(\tilde{p},q_i)} - \tilde{p}q_i\right)} - (1-\tilde{p})
$$

$$
\leq \sqrt{(1-\tilde{p})^2 + 4\tilde{p}(1-q_i)\left(\tilde{p}q_i - \tilde{p}q_i\right)} - (1-\tilde{p}) \tag{161}
$$

$$
= \sqrt{(1-\tilde{p})^2} - (1-\tilde{p}) \tag{162}
$$

$$
= 0. \tag{163}
$$

Thus, $\varepsilon(q_i) = \infty$ for any $q_i \geq 1/(\tilde{r}\tilde{p})$. We may thus restrict our attention to $q_i < \min\{1, 1/(\tilde{r}\tilde{p})\}$.

Since $r^*(\tilde{p}, q_i)$ is a continuous function of $q_i$ and $\varepsilon(q_i)$ is a continuous function of $r^*(\tilde{p}, q_i)$, it follows that $\varepsilon(q_i)$ is a continuous function of $q_i$. By Lemma 2, $\varepsilon(q_i)$ is a decreasing function of $q_i$ for $0 < q_i < \tilde{a}/(\tilde{p}\tilde{r})$ and by Lemma 3, $\varepsilon(q_i)$ is an increasing function of $q_i$ for $\tilde{a}/(\tilde{p}\tilde{r}) < q_i < \min\{1, 1/(\tilde{r}\tilde{p})\}$. Thus, the minimum must occur at $q_i = \tilde{a}/(\tilde{p}\tilde{r})$ and is

$$
\varepsilon\left(\frac{\tilde{a}}{\tilde{p}\tilde{r}}\right) = \log\left(\frac{2\tilde{p}\left(1-\frac{\tilde{a}}{\tilde{p}\tilde{r}}\right)}{\sqrt{(1-\tilde{p})^2+4\tilde{p}\left(1-\frac{\tilde{a}}{\tilde{p}\tilde{r}}\right)\left(\frac{1}{\tilde{r}}-\tilde{p}\frac{\tilde{a}}{\tilde{p}\tilde{r}}\right)}-(1-\tilde{p})}\right) \tag{164}
$$

$$
= \log\left(\frac{2(\tilde{p}\tilde{r}-\tilde{a})}{\sqrt{\tilde{r}^2(1-\tilde{p})^2+4(\tilde{p}\tilde{r}-\tilde{a})(1-\tilde{a})}-\tilde{r}(1-\tilde{p})}\right). \tag{165}
$$

$\square$

**Theorem 5.** *Under the conditions of Theorem 2, let $\tilde{a} < 1$, $\tilde{q} \leq 1$, and $\tilde{r} > 1$, and $0 < p_i < 1$. If the function $r^*$ is such that $r^*(p_i, \tilde{q}) = \max\{\tilde{a}/(p_i\tilde{q}), \tilde{r}\}$ and $r^*(p_i, q_i) = \infty$ for $q_i \neq \tilde{q}$, then the solution to the minimization problem in (13) is*

$$\varepsilon = \begin{cases} \frac{1}{2}\log\left(\frac{\tilde{a}(1-\tilde{q})}{\tilde{q}(1-\tilde{a})}\right), & \text{if } 0 < \tilde{q} \leq \frac{\tilde{a}}{\tilde{r}}; \\ \frac{1}{2}\log\left(\frac{1-\tilde{q}}{\frac{1}{\tilde{r}}-\tilde{q}}\right), & \text{if } 0 < \tilde{q} \leq \frac{1}{\tilde{r}+1} \text{ and } \frac{\tilde{a}}{\tilde{r}} < \tilde{q} < 1; \\ \log\left(\frac{2\tilde{a}(1-\tilde{q})}{\sqrt{(\tilde{r}\tilde{q}-\tilde{a})^2+4\tilde{a}\tilde{q}(1-\tilde{q})(1-\tilde{a})}-(\tilde{r}\tilde{q}-\tilde{a})}\right), & \text{if } \frac{1}{\tilde{r}+1} < \tilde{q} < 1 \text{ and } \frac{\tilde{a}}{\tilde{r}} < \tilde{q} < 1; \\ \log\left(\frac{\tilde{r}-\tilde{a}}{1-\tilde{a}}\right), & \text{if } \tilde{q} = 1. \end{cases} \quad (166)$$

*Proof.* We define $\varepsilon(p_i)$ as follows.

$$\varepsilon(p_i) = \begin{cases} \log\left(\frac{2p_i(1-\tilde{q})}{\sqrt{(1-p_i)^2+4p_i(1-\tilde{q})\left(\frac{1}{r^*(p_i,\tilde{q})}-p_i\tilde{q}\right)}-(1-p_i)}\right), & \text{if } 0 < \tilde{q} < 1; \\ \log\left(\frac{1-p_i}{\frac{1}{r^*(p_i,\tilde{q})}-p_i}\right), & \text{if } \tilde{q} = 1. \end{cases} \quad (167)$$

The solution to the minimization problem in (13) corresponds to the minimum of $\varepsilon(p_i)$ over $(0, 1]$.

We begin with the case where $0 < \tilde{q} \leq \tilde{a}/\tilde{r}$. In this setting, we have that $\tilde{r} \leq \tilde{a}/\tilde{q} \leq \tilde{a}/(p_i\tilde{q})$ so $r^*(p_i, \tilde{q}) = \tilde{a}/(p_i\tilde{q})$ for all $p_i$. By Lemma 2, $\varepsilon(p_i)$ is a decreasing function of $p_i$, so the minimum occurs at $p_i = 1$ and is, by Corollary 2,

$$\varepsilon(1) = \frac{1}{2}\log\left(\frac{1-\tilde{q}}{\frac{\tilde{q}}{\tilde{a}}-\tilde{q}}\right) = \frac{1}{2}\log\left(\frac{\tilde{a}(1-\tilde{q})}{\tilde{q}(1-\tilde{a})}\right). \quad (168)$$

We now consider the case where $\tilde{a}/\tilde{r} < \tilde{q} \leq 1$. Note that $\tilde{r} \geq \tilde{a}/(p_i\tilde{q})$ whenever $p_i \geq \tilde{a}/(\tilde{q}\tilde{r})$. Since in this setting $\tilde{a}/(\tilde{q}\tilde{r}) < 1$, the risk bound can be written in piece-wise form

$$r^*(\tilde{p}, q_i) = \begin{cases} \frac{\tilde{a}}{p_i\tilde{q}}, & \text{if } 0 < p_i < \frac{\tilde{a}}{\tilde{q}\tilde{r}}; \\ \tilde{r}, & \text{if } \frac{\tilde{a}}{\tilde{q}\tilde{r}} \leq p_i < 1. \end{cases} \quad (169)$$

Note that if $p_i \geq 1/(\tilde{r}\tilde{q})$, then $\tilde{r} \geq 1/(p_i\tilde{q})$. Thus, $r^*(p_i, \tilde{q}) \geq \tilde{r} \geq 1/(p_i\tilde{q})$ and so

$$\sqrt{(1 - p_i)^2 + 4p_i(1 - \tilde{q})\left(\frac{1}{r^*(p_i, \tilde{q})} - p_i\tilde{q}\right)} - (1 - p_i)$$
$$\leq \sqrt{(1-p_i)^2 + 4p_i(1-\tilde{q})(p_i\tilde{q} - p_i\tilde{q})} - (1 - p_i) \quad (170)$$
$$= \sqrt{(1-p_i)^2} - (1 - p_i) = 0. \quad (171)$$

Thus, $\varepsilon(p_i) = \infty$ for any $p_i \geq 1/(\tilde{r}\tilde{q})$. We may thus restrict our attention to $p_i < \min\{1, 1/(\tilde{r}\tilde{q})\}$.

Since $r^*(p_i, \tilde{q})$ is a continuous function of $p_i$ and $\varepsilon(p_i)$ is a continuous function of $r^*(p_i, \tilde{q})$, it follows that $\varepsilon(p_i)$ is a continuous function of $p_i$. By Lemma 2, $\varepsilon(p_i)$ is a decreasing function of $p_i$ for $0 < p_i < \tilde{a}/(\tilde{q}\tilde{r})$.

When $0 < \tilde{q} < 1/(\tilde{r}+1)$, note that $1/(\tilde{q}\tilde{r}) > (\tilde{r}+1)/\tilde{r} > 1$ and so $\min\{1, 1/(\tilde{r}\tilde{q})\} = 1$. By Lemma 3, $\varepsilon(p_i)$ is a decreasing function of $p_i$ for $\tilde{a}/(\tilde{q}\tilde{r}) < p_i < 1$. Thus, the minimum must occur at $p_i = 1$ and is, by Corollary 2,

$$\varepsilon(1) = \frac{1}{2}\log\left(\frac{1-\tilde{q}}{\frac{1}{\tilde{r}}-\tilde{q}}\right). \quad (172)$$

When $\tilde{q} = 1/(\tilde{r}+1)$, again $1/(\tilde{q}\tilde{r}) = (\tilde{r}+1)/\tilde{r} > 1$. By Lemma 3, $\varepsilon(p_i)$ is flat for $\tilde{a}/(\tilde{q}\tilde{r}) < p_i < \min\{1, 1/(\tilde{r}\tilde{q})\} = 1$. Thus, the minimum is shared by all points in this range and is equal to (172).

When $1/(\tilde{r}+1) < \tilde{q} < 1$, by Lemma 3, $\varepsilon(p_i)$ is an increasing function of $p_i$ for $\tilde{a}/(\tilde{q}\tilde{r}) < p_i < \min\{1, 1/(\tilde{r}\tilde{q})\}$. Thus, the minimum must occur at $p_i = \tilde{a}/(\tilde{q}\tilde{r})$ and is

$$\varepsilon\left(\frac{\tilde{a}}{\tilde{q}\tilde{r}}\right) = \log\left(\frac{2\frac{\tilde{a}}{\tilde{q}\tilde{r}}(1-\tilde{q})}{\sqrt{(1-\frac{\tilde{a}}{\tilde{q}\tilde{r}})^2 + 4\frac{\tilde{a}}{\tilde{q}\tilde{r}}(1-\tilde{q})\left(\frac{1}{\tilde{r}} - \frac{\tilde{a}}{\tilde{q}\tilde{r}}\tilde{q}\right)} - (1-\frac{\tilde{a}}{\tilde{q}\tilde{r}})}\right) \tag{173}$$

$$= \log\left(\frac{2\tilde{a}(1-\tilde{q})}{\sqrt{(\tilde{q}\tilde{r}-\tilde{a})^2 + 4\tilde{a}\tilde{q}(1-\tilde{q})(1-\tilde{a})} - (\tilde{q}\tilde{r}-\tilde{a})}\right). \tag{174}$$

Finally, when $\tilde{q} = 1$ for $\tilde{a}/(\tilde{q}\tilde{r}) < p_i < \min\{1, 1/(\tilde{r}\tilde{q})\}$,

$$\varepsilon(p_i) = \log\left(\frac{1-p_i}{\frac{1}{\tilde{r}} - p_i}\right). \tag{175}$$

It was shown in (147) that $\varepsilon(p_i)$ is an increasing function of $p_i$ for $\tilde{a}/\tilde{r} < p_i < \min\{1, 1/\tilde{r}\}$. The minimum then must occur at $p_i = \tilde{a}/\tilde{r}$ and is

$$\varepsilon\left(\frac{\tilde{a}}{\tilde{r}}\right) = \log\left(\frac{1-\frac{\tilde{a}}{\tilde{r}}}{\frac{1}{\tilde{r}} - \frac{\tilde{a}}{\tilde{r}}}\right) = \log\left(\frac{\tilde{r}-\tilde{a}}{1-\tilde{a}}\right). \tag{176}$$

$\square$

