# OpenReview forum: "Prior-itizing Privacy: A Bayesian Approach to Setting the Privacy Budget in Differential Privacy"
_NeurIPS.cc/2024/Conference — NeurIPS 2024 poster_

### Official Review · Reviewer_E851 · 2024-07-08

**Soundness:** 2
**Presentation:** 2
**Contribution:** 2
**Rating:** 4
**Confidence:** 4

**Summary:**

For data owners, determining the appropriate privacy level based on the standard interpretation of differential privacy—which limits the likelihood of distinguishability from an adversary's perspective—can be challenging. This paper addresses operational interpretations of (pure) differential privacy by examining the worst-case multiplicative and additive gap between the prior and posterior probabilities of distinguishability when observing the DP mechanism's output. These gaps are defined in terms of relative and absolute disclosure risk. The paper breaks down disclosure risk into two components: a) identifying the presence of an individual's data, and b) identifying certain sensitive features of an individual whose data is present. By contextualizing these two types of disclosure risks, the paper suggests that data owners should create prior-to-posterior risk profiles that reflect their acceptable levels of privacy risk. These risk profiles can then be translated into a pure DP budget by solving an optimization problem or, for a single risk profile, using the provided closed-form solution. The paper includes examples of reasonable risk profiles for specific data-disclosure scenarios and outlines the DP budget required under different prior probabilities of distinguishability by an adversary.

**Strengths:**

- The paper addresses a crucial issue in differential privacy: determining the appropriate DP budget. Examining the gap between prior and posterior probabilities of distinguishability from an adversary's perspective, before and after observing the mechanism's output, is a promising approach.

- Decomposing disclosure risk into the risk of detecting inclusion and the risk of inferring sensitive attributes is valuable for understanding the impact of various DP budgets. This approach underscores that DP can mitigate multiple types of privacy risks, albeit with varying levels of effectiveness.

- The proposed framework provides flexibility in modeling an acceptable disclosure risk profile and includes a closed-form expression for deriving a compliant DP budget from it.

**Weaknesses:**

- The paper suggests shifting the focus from fixing the budget constant $\epsilon$ to fixing a disclosure risk profile (both absolute and relative), which depends on prior risk probabilities $p_i$ and $q_i$. However, this approach complicates the decision-making process for data owners. They either need to define the disclosure risk behavior for an exhaustive range of prior probabilities $p_i, q_i \in [0,1]$, despite the actual population following specific (but unknown) $p_i$ and $q_i$ values, or they must estimate a reasonable range of these prior probabilities for the population (without using their own data) before determining the disclosure risk behavior within that range.

- Decomposing privacy risk into multiple parts, as done in the paper, has certain flaws that have not been considered. This decomposition requires fixing a subgroup $\mathcal{S}$ that is specific to a violation the data curator wants to effectively prevent.
  - Firstly, deriving a DP budget based on one (or a few) choices of such subsets $\mathcal{S}$ is not secure. This is because for every such choice, it is possible to create a pathological mechanism $T^*$ that has a relative disclosure risk of 1 but is $\infty$-DP. Such a mechanism $T^*$ would ensure that the chosen sub-group captured by the subset $\mathcal{S}$ has no influence on the output, but subgroups that haven't been chosen are always revealed.
  - Secondly, for $d$ (binary) features, there are $2^d$ choices of $\mathcal{S}$, and revealing any of them could be considered a privacy violation. It is impractical for a data curator to tailor the privacy risk profile for each of these potential privacy violations to derive the overall DP budget $\epsilon$.
  - Finally, the gradation expressed by first modeling the inclusion bit $I_i \in \\{0,1\\}$ and then the sub-group disclosure bit $Y_i \in \mathcal{S}$ has not been explored thoroughly. By following this chain of thought, it should be possible to define a sequence of increasing specificity, going from the inclusion of a record, to the record being in a sub-population, to the record further being in a sub-sub-population, and so on. The way the $\epsilon$-DP budget affects the prior-posterior gap as we zoom in on a specific record could be crucial and help in deciding the appropriate budget to set.

- It is not clear how the risk profiles considered in Examples 1 and 2 are reasonable and how one can derive such risk profiles for complex situations. The recommendations on choosing a risk profile in lines 229-239 highlight that setting a risk profile can be tricky. A methodology to help set the DP budget should be much more straightforward, which is not evident in these cases.

- The framework is proposed only for $\epsilon$-DP. In practice, $(\epsilon, \delta)$-DP is the more common notion. The paper does not discuss whether and how the framework can be extended to set the $(\epsilon, \delta)$-DP budget.

### Minor Points:

- The assumptions made in the paper, although standard, are incorrectly motivated:
  - Assumption 1, that the output distribution modeled by the adversary using $\mathcal{M}$ matches the actual output distribution of the mechanism, is not to rule out edge-case adversaries exploiting floating point attacks or ignoring the underlying mechanism. It is standard to make Assumption 1 because, in privacy, a data curator needs to protect against the worst-case adversary who knows exactly how the mechanism $T^*$ works, even though a reasonable real-world adversary might not. The rationale is that if the privacy arguments work for such a strong adversary, they will also work for a weaker one.
  - Assumption 2, that $Y_{-i}$ is independent of $Y_i$, is also standard, but not because assuming otherwise would make the adversary considerably stronger. If the data points are independent, but the adversary models them as dependent, the adversary would perform poorly due to the mismatched assumption. It is standard to make this assumption because differential privacy is designed to capture the effect that a single atomic unit of data has on the output distribution. If we assume that $Y_{-i}$ is dependent on $Y_i$, a single atomic unit would be the entire input dataset.

- Lemma 1 is obvious: $\epsilon$-DP with respect to the add or remove operation trivially gives $2\epsilon$-DP with respect to the replacement operation because replacement can be simulated as an addition followed by a removal.

**Questions:**

Following are some questions that would help me understand the presented examples better:

- In Figure 2, deviation of what quantity is being compared?

- When does it make sense to set the relative disclosure risk to $\infty$ as done in equation (16) and (17)?

- In Figure 1, what are the values of $\epsilon_i$ when $q_i \leq 0.1$ where $p_i = 0.05$? The plots in this range seem to be truncated.

**Limitations:**

The paper has several limitations regarding applicability of the techniques which haven't been discussed. I suggest the authors to shed some light on the following limitations:

- How to choose the disclosure subsets $\mathcal{S}$ without knowing what a potential adversary might be interested in inferring?
- How to decide the appropriate $\epsilon$ from the disclosure risk profile as plotted in Figure 1? For certain choices of $p_i, q_i$, the values of $\epsilon$ could be excessively lax. On the other hand, if we choose the smallest $\epsilon$ pessimistically, the value might be very close (or identical) to the trivial solution equation (7)
- What are the guidelines that may help a data owner in deciding an appropriate disclosure risk-profile that caters to their specific setting?
- What is a fail-safe disclosure-risk profile that should be included with the other risk profiles to ensure that blatant non-privacy never happens?
- Can the framework be extended to $(\epsilon, \delta)$-DP?

---

> ### Author Rebuttal · Authors · 2024-08-06
>
> Thank you for the thoughtful comments. To frame our response, we begin with two general points. First, our method is intended for the standard definition of DP, and we presume the agency will apply a DP algorithm with the selected epsilon. Thus, the DP guarantee associated with the selected epsilon applies to all subgroups, not only the group that satisfies the particular $\mathcal{S}$.
>
> Second, our work is an initial paper on determining an appropriate privacy budget for a given data release; there is more to be done. Based on our review of the literature, practitioners often have difficulty understanding how to set an epsilon that provides a satisfactory balance of risk and utility (see line 20 of the text). Yet decision-makers in statistical agencies have decades of experience interpreting disclosure risk measures when establishing (legacy) confidentiality protection methods. By linking formal privacy guarantees to these concepts, we intend to provide decision-makers familiar with statistical risk measures tools to assess risk-utility trade-offs of different epsilon.
>
> Weaknesses
> - We agree that implementation details may be complex, as can be the case for engineering DP solutions in general. Whether working with risk profiles is less complicated for decision-makers than reasoning about the risk-utility trade-offs for different epsilon using some other ad hoc heuristic is a matter of opinion, which may differ by decision-maker or problem setting. Our intention is to provide a framework for additional paths to setting epsilon that can be developed further to accommodate specific implementations.
> - The privacy guarantee comes from using a DP algorithm with the selected epsilon. The $\mathcal{S}$ (henceforth, S) is a tool for interpretation and does not affect the guarantee (other than through influencing the choice of epsilon), which applies for all individuals in the data. Of course, tuning epsilon based on a specific S and utility evaluations may result in a looser or tighter guarantee than if the agency were to consider other S or utility evaluations. This seems inevitable with any method for selecting epsilon that considers the risk-utility trade-off. For example, the Census Bureau relied on specific reconstruction attack success rates and utility metrics to decide the ultimate privacy budget (and algorithm itself) for the 2020 census products. The final product may have changed with other disclosure risk or utility evaluations. Nonetheless, the reviewer’s point suggests that the agency should include in S any secrets it deems critical to protect. This need not be every possible S—as the reviewer notes, this is impractical for any method that seeks to manage the risk and utility trade-off—but it could involve evaluations of multiple risk profiles. The question then, which we did not consider in this initial paper, is how the agency should select among them if they present different recommendations. We thank the reviewer for raising this issue and will revise the paper to point to it as future research.
> - We agree that some agencies’ risk profiles may be more complex than those in the text. Our goal is to lay out a general framework, with the expectation that researchers and practitioners could tailor implementations for their particular settings; we leave these developments to future work. That said, the literature indicates that practitioners presently do not understand how to set epsilon to ensure desirable risk-utility trade-offs. Given the lack of standards, even simple risk profiles can be helpful for managing and understanding the trade-offs.
> - See our response to reviewer HWnJ, where we discuss the extension to approximate DP.
>
> Minor Points
> - Thank you for this perspective on our assumptions. We are updating their presentation accordingly.
>
> Questions
> - In Fig 2, to avoid privacy leakage we consider hypothetical deviations of 0.25, 0.5, 1, and 2 from the target of 6.0. See the text beginning at line 268. To make this clearer, we are adding a note to the caption.
> - The aim in selecting risk profiles of this form is to simplify the analysis. In Ex 1, the agency focuses on a class of adversaries who already know the target’s demographic information; such adversaries have $q_i = 1$. Operationally, the bound on the relative risk is set to $\infty$ for $q_i \neq 1$ in (16) to represent that these adversaries are outside the analysis. Ex 2 and (17) employ similar logic with a different class of adversaries. We do not suggest setting $\epsilon = \infty$. We are updating the text to clarify this point.
> - In Fig 1, $\epsilon_i$ is large for small $q_i$, e.g., $\epsilon_i = 5.4$ when $q_i = 0.01$. We truncate values >3 for readability. We are updating the caption to clarify.
>
> Limitations
> - The agency can decide which S it considers particularly sensitive, e.g., it may consider whether an individual has a disease more sensitive than their age. When multiple S are of interest, the agency has a decision problem on its hands. As with designing DP solutions in general, the agency must prioritize some of these S over others, e.g., using decision-theoretic criteria. This is an important topic for future work. We thank the reviewer for pointing it out and are updating the text to highlight this topic.
> - We take the minimum epsilon as the recommendation (see (13)). This could be a pessimistic choice although, as discussed in our response to reviewer QVWW, there still can be large gains over the baseline recommendation.
> - We envision agencies could determine risk profiles analogously to the solicitation of utility functions in decision theory, e.g., by considering a series of bets. Since our work is a first effort at defining a framework, we leave this development to future work and update the text accordingly.
> - There is no risk of blatant non-privacy since the release satisfies DP with a finite epsilon.
> - See our response to reviewer HWnJ, where we discuss the extension to approximate DP.

---

> > ### Comment · Reviewer_E851 · 2024-08-12
> >
> > I appreciate the responses and I agree that we need interpretations of DP that allows setting the parameters appropriately.
> >
> > > Yet decision-makers in statistical agencies have decades of experience interpreting disclosure risk measures when establishing (legacy) confidentiality protection methods.
> >
> > Decision-makers rely on academics for deciding on the risk measurement and appropriate budget to set in my experience. There is rarely a clear consensus on risk measurements despite the decades of involvement in such decisions.
> >
> > While the framework proposed is a nascent attempt at solving a complex problem, the complexity of the approach in general is a considerable drawback.

---

> > > ### Author Response · Authors · 2024-08-12
> > >
> > > Thank you for taking the time to read and respond to our rebuttal.
> > >
> > > We appreciate your comment that setting epsilon in this (or any) principled way can be complex. We imagine you might concur that the whole process of implementing formal privacy solutions in genuine contexts is a highly complex endeavor.
> > >
> > > That said, the literature on setting epsilon in DP in a formal manner is underdeveloped. Despite its importance, there are many settings where we are aware of no existing method for this task (prior to our work). We are optimistic that our approach offers a framework that can be further developed to provide options for addressing this gap in the literature.
> > >
> > > Thank you again for your detailed feedback throughout the review process. We appreciate the time you devoted to our work.

---

### Official Review · Reviewer_HWnJ · 2024-07-12

**Soundness:** 3
**Presentation:** 3
**Contribution:** 3
**Rating:** 7
**Confidence:** 2

**Summary:**

This paper proposes a bayesian framework for determining the DP budget \eps. In particular, the authors develop a mathematical technique based on how much posterior risk the agencies are willing to accept given some prior risk and the \eps obtained through their formulation is unique.

**Strengths:**

Although there has been much debate on the topic of how to determine privacy budget and different flavors of this formulation have been proposed in prior works, I have not seen this formulation in the context of agencies deciding their \eps privacy budget. Authors also do a great job in differentiating their work to prior works that have proposed similar ideas. The ideas in this paper seem novel and useful to real world scenarios.

**Weaknesses:**

This paper only discusses the setting of \eps in the context of pure DP. It does not discuss whether these techniques could be extended to weaker notions of DP such as approx DP. Since in most real world applications agencies use approx DP notions I feel like this is an important discussion point that is currently missing.

**Questions:**

How do your methods extend to weaker notions of DP? (see comment in weaknesses)

**Limitations:**

See weaknesses comment.

---

> ### Author Rebuttal · Authors · 2024-08-06
>
> Thank you for raising this point. A few works such as Kasivisiwanathan & Smith (2014) and Kifer et al. (2022) discuss Bayesian semantics of approximate DP, although the details differ from the semantic characterization we use in this work for pure DP. We imagine one could use these results as the basis for a similar framework for approximate DP, following our framework as a blueprint. We did some initial investigations of this and found the generalization not to be straightforward. Since our work is a first effort on formalizing the selection of $\varepsilon$ by linking it to statistical disclosure risk profiles, we opted to focus on the pure DP setting and leave this extension for future work. But we agree with the reviewer about the significance of this extension and are updating the text to highlight that it is an important open problem, and we suggest some possible first steps for that future research.

---

### Official Review · Reviewer_QVWW · 2024-07-13

**Soundness:** 3
**Presentation:** 3
**Contribution:** 2
**Rating:** 6
**Confidence:** 4

**Summary:**

This paper proposes a novel method for selecting epsilon that comes with a natural adversarial interpretation. They characterize an adversary’s auxiliary knowledge by two prior probabilities: the prior belief that an individual participated in the dataset, and the prior belief that an individual’s record has some characteristic. Based on these two parameters, an institution can choose a “risk profile”, that maps an adversary’s prior knowledge to their allowable increased posterior knowledge after viewing the DP output. The authors propose a mapping from risk profile to maximum allowable epsilon.

A major insight is that risk profiles which allow adversaries with weak priors to learn more than those with strong priors allow for larger epsilon values than a constant profile would allow (constant bound on posterior/prior ratio).

The authors go on to provide instructive examples of risk profiles that different institutions may want, and the resulting epsilon bounds. They include an analytic experiment based on Durham infant mortality rate data.

**Strengths:**

This paper addresses the two primary limitations of differential privacy at once: 1) an interpretable way of setting the privacy loss parameter, and 2) an analytic justification of higher epsilon values.

The paper writing is strong: their framework is laid out very clearly, and the examples and guidelines are useful and clarifying. The appendix is thorough and supplements most content that I wanted to see added.

Of course, much ink has been spilled over adversarial interpretations of epsilon and different adversarial threat models (especially Bayesian ones). I felt that the authors did their homework, and nicely positioned their work in the literature.

**Weaknesses:**

The two main weaknesses that I noticed revolve around baseline clarity:
Little is offered in the main paper to show improvements over the naive baseline ($\epsilon = \log(r)/2$). I noticed a note on this after Figure 2, but felt it should be highlighted more. The improvement in utility over the baseline $\tilde{a} = 0$ is nonzero, but not enormous. The question is whether that improvement warrants the increased complexity of understanding the $\tilde{a} > 0$ regime (i.e. the complexity of choosing a non-constant risk curve).
A more thorough exploration of risk profiles and the above baseline. Take a risk profile with minimum $r’$. Set its baseline to the constant risk profile with value $r’$. Can we say anything about the difference between their epsilons? Are there settings where this gap is very large? If so, are these settings realistic?

I apologize if I missed the above addressed in the paper, but to me this is the most critical question at hand. If we can’t make statements about this gap, then we don’t know how much we can gain over the baseline.

**Questions:**

See weaknesses section.

**Limitations:**

The paper identifies its assumptions early on. I think the major limitation is the relatively small boost in epsilon value (and therefore utility) for the risk curves demonstrated.

---

> ### Author Rebuttal · Authors · 2024-08-06
>
> Thank you for the useful feedback regarding baseline clarity. On the question of demonstrating improvements over the baseline, the extended versions of Examples 1 and 2 in the supplement demonstrate the potential for substantial improvements over the baseline. For example, Table 4 and the accompanying discussion (lines 862-866) show an increase by more than a factor of two in the recommended $\varepsilon$ in all cases. We agree that this comparison is an important aspect of our contribution and are updating the discussion of the examples in the main text to highlight this improvement.
>
> The second point you raise is quite thought-provoking and not one we had previously considered. The gap between the baseline and our recommendation can be large. As an illustrative example, suppose an agency has a simple, “point” risk profile, where for $p_i = 0.5$ and $q_i = 1$, they desire a relative risk bound of $r’ < 2$. That is, $r^*(0.5, 1) = r’$. The baseline recommends $\epsilon = \log(r’)/2 < 0.35$. The recommendation from Theorem 2 can be shown to have the form $\epsilon = \log(r’/(2 – r’))$. This diverges as $r’ \to 2$. Thus, the gap between the baseline and the recommendation from Theorem 2 can be arbitrarily large for $r’$ sufficiently close to $2$.
>
> The example above likely does not represent a realistic disclosure risk profile of a real-world agency, but it is instructive in determining when our method can outperform the baseline. In general, the baseline’s recommendation is smaller because it is necessarily enforcing a relative risk of $r’$ for adversaries with small priors. If $r’=2$, then an adversary with prior probability $p_i q_i = 0.001$ is allowed a posterior probability of at most $0.002$, which is very restrictive and leads to a small $\varepsilon$ recommendation. If adversaries with small priors are ignored - as in the above point example - or allowed to have large relative risks - as in the extended versions of Examples 1 and 2 - then the recommendation from our method will outperform the baseline, and possibly by a substantial amount.
>
> We agree that a more thoughtful discussion around these points provides useful context to our examples and are adding the main points we discuss above to the paper.

---

> > ### Comment · Reviewer_QVWW · 2024-08-13
> >
> > Thank you for addressing my concerns. I've revised my score from 4 to 6.
> >
> > I want to emphasize that *the* value of adopting this methodology is the promise of an increased DP epsilon, and thereby improved utility. With the authors agreeing to update the discussion of the examples in the main text to highlight improvement over the baseline, I have decided to raise my score.
> >
> > The above comment on the possibility of arbitrary improvement over the baseline is interesting. A more complete analysis on when semantically meaningful profiles make arbitrary (or large) improvements on epsilon would encourage me to improve my score further.
> >
> > I missed the fact that Table 4 (implicitly) shows a 2x improvement of epsilon, which does feel significant. However, the fact that the table shows $\tilde{r}$ from which I need to derive the baseline (naive) $\epsilon = \log(\tilde{r})/2$ to see the improvement needs to be fixed. The fact that this most central point -- improvement of epsilon over the naive bound -- can only be numerically derived from a table deep in the appendix (page 25) is poor presentation that I hope the authors will improve in the main paper.
> >
> > Thank you again for the above clarified points

---

> > > ### Author Response · Authors · 2024-08-13
> > >
> > > We appreciate your willingness to raise your score and your helpful comments on our work.  As you suggest, we will prominently emphasize the increase in $\varepsilon$ from our method generally and, in particular, clarify the gains in $\varepsilon$ from Table 4 as we update the main text.  We also appreciate your suggestion of further analysis of when/why risk profiles admit increased $\varepsilon$.  This is something we plan to investigate as part of our future work in developing this framework.  Thank you for the research topic suggestion and the fruitful conversation.

---

### Official Review · Reviewer_XDrZ · 2024-07-13

**Soundness:** 3
**Presentation:** 3
**Contribution:** 3
**Rating:** 6
**Confidence:** 3

**Summary:**

The paper proposes a novel framework for setting an appropriate privacy budget via controlling Bayesian posterior probabilities of disclosure. The connection is established through the risk profile, an upper bound on disclosure risk involving privacy parameters. Theoretical justification and empirical evaluation are also provided to support this framework.

**Strengths:**

* The paper links differential privacy to Bayesian disclosure risk through the risk profile, providing a reasonable framework for selecting privacy parameters.
* The paper provides some ready-to-use tools for implementing this framework, supported by empirical results.
* The paper is well-written, and the authors provide a comprehensive discussion on its connection to related work.

**Weaknesses:**

* The framework is designed for releasing discrete statistics, and it is unclear whether it can be extended to release continuous statistics.

**Questions:**

Can the framework proposed in this paper be extended to handle cases where $T(Y)$ has continuous support?

**Limitations:**

The authors address this part in the limitations section of the checklist.

---

> ### Author Rebuttal · Authors · 2024-08-06
>
> Thank you for noting this omission. Generalization of our results from the discrete case to the continuous case can be accomplished by replacing sums with integrals and probability mass functions with probability density functions throughout the theorem statements and proofs. We focus on the discrete case throughout the document for clarity of presentation. We apologize that this was not clear in the original manuscript and are updating the text to highlight this point.

---

> > ### Comment · Reviewer_XDrZ · 2024-08-11
> > **Official comment by reviewer XDrZ**
> >
> > Thank you for answering my question. I will keep my positive score.

---

### Decision · Program_Chairs · 2024-09-25

**Decision:**

Accept (poster)

**Comment:**

This paper proposes an approach to setting the privacy budget $\epsilon$ in (pure) differential privacy, based on an organisation’s appetite for (Bayesian) posterior risk of disclosure for given level of prior risk of disclosure. Specifically, the approach uses a so-called ‘risk profile’ which upper bounds disclosure risk. Disclosure risk is decomposed into: identifying presence of an individual; and identifying sensitive features. A strawperson policy of a constant profile (constant bound on posterior/prior ratio) is improved by allowing adversaries with weak priors to learn more (have larger $\epsilon$) than those with strong priors.

Reviews raise concerns about: discrete vs continuous responses (authors clarify that discrete is used for ease of exposition; while it would be ideal to highlight this in the paper, I believe there should be a qualifier that due to limitations of fixed-precision floating-point, that discrete responses are preferred in practice); relatively limited improvement of utility over baseline in the body (however enabling of a doubling of $\epsilon$ in the supplement; accordingly the reviewer raised their score – reflecting the main benefit of the work not per se being interpretable $\epsilon$ but unlocking improved utility); requests for a more thorough discussion of risk profiles (the authors provide a thoughtful reflection in their response that would improve the paper if added); that the framework applies only to pure DP with extension to the much more common approximate DP not known or seemingly available (a residual concern for the work’s impact); that setting risk profiles may not actually be easier than setting a single $\epsilon$; concerns raised on the impact of decomposing privacy risk into two components, that could lead to flaws exploitable by certain pathological mechanisms (demonstrated by E851) were addressed by the authors due to the choice of mechanism’s DP guarantee holding for all subgroups $S$; and (fixable) incorrect motivations/explanations of Assumptions 1 and 2 (corrected by E851).

Setting of DP’s privacy parameters is a well-known policy challenge, with a number of proposals available in the literature. While a major drawback – of a framework intended for use by policy makers – is the complexity of the approach, the framework is nonetheless interesting. This is an important direction. The paper’s approach is contextualised in the literature, is novel, and connects to ideas of Bayesian disclosure risk from statistical disclosure control. The paper is well written, with pedagogical examples of risk profiles, scenarios, providing the reader with helpful reference points to use the approach.